# A Hybrid POI Recommendation System Combining Link Analysis and Collaborative Filtering Based on Various Visiting Behaviors

Sumet Darapisut, Komate Amphawan *, Nutthanon Leelathakul  and Sunisa Rimcharoen 

Faculty of Informatics, Burapha University, Chonburi 20131, Thailand; dearsumet@gmail.com (S.D.);
nutthanon@buu.ac.th (N.L.); rsunisa@buu.ac.th (S.R.)
* Correspondence: komate@gmail.com

**Abstract:** Location-based recommender systems (LBRSs) have exhibited significant potential in providing personalized recommendations based on the user's geographic location and contextual factors such as time, personal preference, and location categories. However, several challenges (such as data sparsity, the cold-start problem, and tedium problem) need to be addressed to develop more effective LBRSs. In this paper, we propose a novel POI recommendation system, called LACF-Rec3, which employs a hybrid approach of link analysis (HITS-3) and collaborative filtering (CF-3) based on three visiting behaviors: frequency, variety, and repetition. HITS-3 identifies distinctive POIs based on user- and POI-visit patterns, ranks them accordingly, and recommends them to cold-start users. For existing users, CF-3 utilizes collaborative filtering based on their previous check-in history and POI distinctive aspects. Our experimental results conducted on a Foursquare dataset demonstrate that LACF-Rec3 outperforms prior methods in terms of recommendation accuracy, ranking precision, and matching ratio. In addition, LACF-Rec3 effectively solves the challenges of data sparsity, the cold-start issue, and tedium problems for cold-start and existing users. These findings highlight the potential of LACF-Rec3 as a promising solution to the challenges encountered by LBRS.

**Keywords:** point-of-interest recommendations; link analysis; collaborative filtering; distinctiveness

## 1. Introduction

As location-based data from mobile devices and social networks are increasingly available, location-based recommender systems (LBRSs) [1–7] hold significant promise in providing personalized recommendations to users based on their locations and contextual factors. LBRSs recommend points of interest (POIs) such as restaurants, museums, and shopping centers to users after considering the users' current location, historical check-in records, social context, and other relevant information. The primary objective of LBRSs is to facilitate users in discovering captivating venues, improving their user experience, and contributing to the growth of local businesses and tourism.

LBRS have attracted significant attention from researchers and industry professionals due to their potential implications across diverse domains including tourism, transportation, and marketing. In tourism, for instance, LBRSs help visitors explore novel and captivating attractions, events, and restaurants within unfamiliar cities. In transportation, LBRSs can assist commuters and travelers plan their journeys and identify convenient and efficient routes. In marketing, LBRSs can facilitate enterprises in promoting their products and services to potential customers based on their location and preferences.

Numerous location-based social network systems (LBSN) platforms including Foursquare, Yelp, and Facebook Places utilize POI recommendation systems to overcome the issue of information overload by identifying interesting venues and filtering out irrelevant options. From the enterprise perspective, these POI recommendation systems can yield substantial profit for proprietors because users can perceive their preferred locales, thereby augmenting user loyalty. Nonetheless, there are several challenges to POI recommender systems.

The first challenge lies in the issue of data sparsity. In real-word scenarios, the number of locations is growing rapidly, however, users tend to check-in only a few times within their immediate vicinity. Consequently, the user–POI matrix becomes sparse, causing difficulties in determining the users' preferences.

The second challenge is the cold-start problem encountered when users, who seek location recommendations (referred to as target users), lack a history of check-ins. In particular, traditional approaches rely on the similarity between places visited by the neighbors of users with existing check-in histories and those visited by target users. Hence, if the target users are new, the LBRS struggles to effectively capture their location preferences.

To address these two challenges, link analysis-based (LA) algorithms such as the HITS (Hypertext Induced Topic Search) model [5,8,9] rely on graph techniques to extract quality POIs and local experts. These algorithms can leverage the knowledge and the experience of local experts within local areas (of which the associated data are non-sparse) to recommend potential locations for users without a history of check-ins (referred to as cold-start users). However, if there is a sufficient amount of user check-in data, CF methods are likely to outperform LA methods [10,11].

Collaborative filtering (CF) algorithms [6,12,13] rely on a user's check-in history to provide more accurate recommendations, especially when the number of check-ins of the user has increased. Nevertheless, in the case of the cold-start problem, LA algorithms tend to surpass CF algorithms [5,14,15].

The third challenge involves the tedium problem, where the majority of CF and LA algorithms rely only on the check-in frequency, often leading to the recommendation of predominantly popular places without considering personal preference. This tendency arises because the algorithms do not consider location diversity and novelty.

To tackle all three challenges, we propose a novel POI recommendation system called LACF-Rec3. This system utilizes a hybrid method combining link analysis and collaborative filtering. It is based on three visiting behaviors: frequency, variety, and repetition. LACF-Rec3 identifies interesting and distinctive POIs by analyzing the user- and POI-visit patterns, subsequently ranking them, and delivering recommendations to cold-start users. For users with existing check-in histories (referred to as existing users), LACF-Rec3 performs collaborative filtering based on their check-in history and the distinctive aspect of the POIs.

We finetuned a HITS-based model and called it HITS-3 (HITS based on three check-in behaviors) to discover interesting locales with distinctiveness (i.e., distinctive aspect) in all three location characteristics: visit frequency, user variety, and repeated check-ins. Locations with high check-in frequencies typically denote popular or trending destinations with a high number of visits. High-variety locations signify places visited by various individuals including tourist attractions and landmarks that draw a varied audience without recurring visits. Locations with the great number of repeat check-ins could be routine daily venues such as grocery stores or supermarkets as well as specific places tailored to particular users such as gyms, basketball courts, or board game cafes.

Furthermore, our HITS-3 model also discovers places with distinctiveness in all three user characteristics: check-in frequency, location variety, and the number of locations the user revisits. Users exhibiting high check-in frequency epitomize active and sociable individuals who frequently engage in location-based activities. Those with high location variety demonstrate an affinity for novel experiences and avoid revisiting the same places frequently. Users with a high number of locations revisited often possess specific preferences or display strong loyalty as dedicated patrons of specific locations.

Derived from the aforementioned HITS-based model, the distinctiveness and distinctive scores (explained in Section 3) of all locations and all users could be utilized to recommend locations for cold-start users. For target users with check-in histories, locations are recommended based on the interest scores calculated by our enhanced user-based collaborative filtering algorithm, called CF-3 (Collaborative Filtering based on 3-distinctiveness). Specifically, when a user already has a check-in history, CF-3 suggests locations based on

other users who exhibit similar distinctiveness and distinctive scores. Subsequently, CF-3 selects only interesting POIs that are in the vicinity of the target user's current location.

In this paper, we present the design and implementation of LACF-Rec3 (a combination of the HITS-3 and CF-3 methods), followed by a performance evaluation using a Foursquare dataset. Our experiments reveal that LACF-Rec3 outperforms the previous methods in terms of recommendation accuracy, ranking precision, and matching ratio. These results emphasize the potential of LACF-Rec3 as an effective solution to the challenges confronting LBRS.

In summary, the four main contributions of our novel POI recommendation system (LACF-Rec3) are as follows:

- We introduce a pioneering concept for capturing visiting behavior by considering three key characteristics, namely frequency, variety, and repetition. These are utilized to discover intriguing places and identify the distinctiveness of the POIs and users based on their visiting patterns.
- LACF-Rec3 aims to provide captivating POIs for both cold-start and existing users. Our LACF-Rec3 combines a new extended version of the link analysis approach with a novel collaborative filtering recommendation system.
  a. We introduce HITS-3, which is an extended version of the link analysis recommendation technique. It integrates a HITS-based model and considers the distinctiveness of both users and POIs to generate the top-ranked POIs specifically for cold-start users.
  b. We also propose a novel collaborative filtering technique, called CF-3, that considers the similarity in the users' distinctiveness to provide highly personalized recommendations for existing users.
- In addition, we propose a new metric, termed "Matching Ratio". The evaluation of LBRS performance typically relies on standard metrics such as precision and recall. Ideally, they should have been determined by comparing the list of recommended places with the list of locations where users have checked in after reviewing the recommendation. However, in the context of experiments, the locations suggested by the LBRS are not actually presented to users for selection. Hence, precision and recall are typically computed by comparing the list of recommended places with the list of checked-in locations, even though the users have not engaged with the recommended list. As a result, relying solely on precision and recall may not accurately capture the LBRS's recommendation accuracy. To overcome this limitation, we propose a novel evaluation metric that measures the matching between the recommended locations and the preferences of the target users. This new metric supplements the precision and recall measures, thereby enhancing the overall evaluation capabilities.
- We conduct experimental evaluations using a real-world Foursquare dataset. Our LACF-Rec3 method significantly outperforms other methods in terms of the recommendation accuracy and ranking accuracy, as demonstrated by the extensive evaluation experiment results in terms of precision, recall, and NDCG (normalized discounted cumulative gain) [16] metrics.

The remainder of this paper is structured as follows. Section 2 provides an overview of the related work in LBRSs and outlines the main approaches and techniques employed in the current literature. Section 3 describes our proposed approach in detail. In Section 4, we provide insight into the experimental setup, the evaluation metrics, and the results of the experiments conducted on the real-world dataset. Section 5 discusses the results, the limitations, and the future directions of the proposed approach.

## 2. Related Work

Our proposed method generates a POI recommendation list for a cold-start user and an existing user based on the link-analysis technique and collaborative-filtering technique,

hence in this section, we review the existing POI recommendation systems that take advantage of the link-analysis technique and collaborative-filtering technique.

### 2.1. Link-Analysis Technique

Link-analysis techniques play a crucial role in recommender systems by leveraging the connections and relationships between users, items, or other contexts. In particular, a HITS-based model [8,17–19] is one of the link-analysis algorithms that is widely used in generating POI recommendations. The HITS-based model is a fundamental approach, designed to assess the significance of webpages interlinked to each other. The HITS algorithm assesses the authority scores and hub scores of web pages. The former measures the relevance and importance of a page's topics, while the latter represents a page's ability to link to other authority pages on the same topic. The algorithm iteratively computes both scores. An authority score of a web page, $P$, is the sum of the hub scores of the pages pointing to $P$, while $P'$ s hub score is the sum of the authority scores of the pages $P$ points to. In the context of LBSN, the scores of the POIs are determined in a manner analogous to the calculation of authority scores, while those of the users are computed using a method similar to the method for hub scores. For example, the first study of the application of the HITS algorithm to generate a POI recommendation list, called the tree-based hierarchical graph (TBHG), was presented by Zheng et al. [20]. The hub nodes according to the graph structure in the HITS algorithm are the users, and the authority nodes are groups of GPSs that can be addressed as locations. Bao et al. [5] proposed the location-based and preference-aware recommendation method. This approach generates recommendations for places by incorporating collaborative filtering and the HITS algorithm. The weight category hierarchy (WCH) is considered, which involves analyzing the frequency of the visited categories. Once experts in the area are identified, the WCH is further used to find similarities and generate a list of interesting recommendations for the target users. Long and Joshi [14] presented the HITS-based POI recommendation method. This approach improves the HITS algorithm by considering the diversity of check-ins by using entropy, based on the assumption that users with diverse check-ins may indicate their expertise in the area. In addition, this method takes into account the relationships with the user's friends in the social network. Bagci and Karagoz [21] presented Context-aware Location Recommendation with Random Walk method (CLoRW) to generate personalized POI recommendations. This method improves the random walk algorithm by taking into account local experts using traditional HITS methods, POI, the user's current POI, friendship relationship, POI popularity, etc. Ying et al. [15] proposed an approach to generate personalized POI recommendations, and aimed to address the sparsity problem by considering category-based replacement of locations in the CTD (context-aware tensor decomposition) process and by identifying interesting POIs using the WHBPR (weighted HITS-based POI rating) step with the additional consideration of friend relationships. In 2020, an N-most interesting location-based recommender system called NILR [9] was presented to generate a POI recommendation list for a cold-start user. The NILR considered both the frequency of visits and the user's preferences (i.e., the number of locations the user revisits). Then, the ranking procedure was applied to generate a final recommendation list. Sun et al. [22] presented a weighted HITS-based model algorithm to generate POI recommendation lists. This method recommends interesting places based on check-in frequency, number of transit points to the location, and time interval to visit in each location obtained by the improved HITS-based model to all active users (without considering their preferences). Yin et al. [13] proposed a tensor decomposition based collaborative filtering (TDCF) algorithm. The TDCF uses the tensor decomposition structure to consider the relationship of the users and check-in place categories along with the check-in interval of a user. The approach also aims to solve the sparsity problem by filling in the missing check-ins using the tensor decomposition technique. Then, the popularity of the location is determined using the HITS algorithm. The algorithm recommends popular locations to the user if they are close to the user's current location. Recently, a privacy-preserving

time-aware recommendation (PPTA-RM) technique was introduced [23]. This technique incorporated both coarse-grained and fine-grained recommendations to anticipate where users might go in the upcoming time slot. For the coarse-grained level, the method captured the users' preferences for POI categories using an extended matrix factorization technique and predicted the preferences using singular spectrum analysis (SSA). At the fine-grained level, a preferred location was discovered using an improved version of the hyperlink-induced topic search (HITS) algorithm.

### 2.2. Collaborative Filtering Technique

A collaborative filtering (CF) technique [24–26] is a technique in recommender systems for generating personalized recommendation lists in various domains such as places, movies, songs, news, videos, and events. It can be divided into two primary categories: model-based and memory-based techniques. Model-based CF techniques [26–28] employ mathematical models to learn and predict the user preferences. For example, matrix factorization was used in [26] to recognize latent patterns within user behavior and item characteristics, facilitating personalized recommendations. The matrix factorization techniques use vectors in a latent space to represent users and items, and decomposes the user–item interaction matrix into latent factors, thereby enabling the discovery of hidden patterns in user preferences and item characteristics. On the other hand, the memory-based CF techniques utilize the complete user–item interaction dataset to generate recommendations. By evaluating similarities between the users or items, these techniques identify the closest neighbors and offer personalized recommendations aligned with the user preferences. The memory-based CF technique can be further divided into user-based CF and item-based CF. In user-based CF, the systems [25] actively scan for other users with similar behaviors based on the historical data of the target user and generate recommendation lists. Similarly, item-based CF algorithms [24] search for similar items based on the consumption history of the target item and generates recommendation lists.

Earlier works involved the applications of the CF technique for location-based social networks (LBSNs) are as follows. Baral and Li [27] introduced two combined models for personalized POI recommendation: a ranking-based model and a matrix factorization-based model. They incorporated crucial factors such as visit frequency, social connections, time, location, and categorization into a unified recommendation framework. Zhao et al. [28] proposed a Sentimental-Spatial POI Mining (SPM) method by fusing sentimental and geographical attributes of locations. This work also proposed a Sentimental-Spatial POI Recommendation (SPR) model for personalized recommendations by considering sentiment similarity and geographical distance factors based on matrix factorization to generate a personalized location recommendation list. Yuan et al. [29] proposed a time-aware point-of-interest (POI) recommendation model by improving the user-based CF method. This method considered temporal and geographic information for the time context and solved the sparsity problem by using the smoothing technique to generate efficient POI recommendations. Si et al. [30] introduced an adaptive POI recommendation method called CTF-ARA by considering temporal features and user activity. Users are classified into active and inactive users based on their activity. For inactive users, the popularity of POIs is considered to find the similarity of users based on all time slots to generate POI recommendation lists. For active users, this method considers POI popularity based on sequential time slots to increase the recommendation accuracy of the list of recommended places for users. In 2019, a memory-based POI preference attenuation model algorithm was proposed by Gan and Gao [12]. This approach generated a list of personalized location recommendations for users based on the collaborative filtering method. This method determined the similarity of the user's check-in behavior with other users in the system and increased the importance of the user's recently visited locations using the Ebbinghaus forgetting curve technique. Khazaei and Alimohammadi [31] proposed a context-aware group-oriented location recommendation system (CLGRW) for LBSNs based on a random walk algorithm. CLGRW considers user contexts (e.g., social relationships, personal preferences), location

contexts (e.g., category, popularity, capacity, and spatial proximity), and environmental contexts (e.g., weather, day of the week). These contexts are based on a random walk with restart (RWR) algorithm. Zhang et al. [32] introduced the POI recommendation framework using the users' memory-based preferences and the POI stickiness method (U-CF-Memory-Stickiness), which is an improved version of the memory-based POI preference attenuation model. The U-CF-Memory-Stickiness assigns high scores to locations that the user has recently visited using the Ebbinghaus Forgetting Curve engine. In addition, this method also considers the revisit location with the POI stickiness method based on the collaborative filtering method. Recently, a CULT-TF method [33] was introduced to generate a personalized POI recommendation list. The method combined the contextual information of similar users into the tensor factorization model. A user clustering method was proposed to select active users with the greatest impact and influence. A U-L-T tensor was also proposed as the basis for creating a POI recommender system by considering the user activity, POI popularity, and time slot popularity.

### 2.3. Comparative Analysis of Baseline Approaches and Our Proposed Method (LACF-Rec3)

The aforementioned research works have provided POI recommendation lists primarily based on the link analysis technique and/or collaborative filtering technique. However, various earlier studies focused only on utilizing frequency. As a result, these location recommendations may suffer from a tedium problem, where locations are chosen predominantly based on their popularity, and not their variety and repetition rates. Therefore, we proposed the concept of location and user distinctiveness, in terms of frequency, variety, and repetition, to capture a broader spectrum of user preferences. In addition, we aimed to provide a tailored recommendation list for both cold-start and existing users by effectively coping with the cold-start and sparsity issues. For cold-start users, interesting locations are recommended by the proposed HITS-3 method, based on the distinctiveness of locations in their vicinity. For existing users, interesting venues are recommended by the CF-3 method.

Table 1 illustrates the comparative analysis between the baseline approaches and our proposed method. We observed that there was one prior research work [14] that recommended locations based only on user variety. Additionally, two previous research works [9,32] considered the check-in repetition. Prior hybrid recommendation systems [5,13,15,23] that had utilized both the link analysis model (i.e., HITS based model) and the collaborative filtering approach were utilized. However, there have been no research works that considered three check-in behaviors (i.e., frequency, variety, and repetition) while taking advantage of the distinctiveness of the users and locations to generate a POI recommendation list.

**Table 1.** Comparative analysis of the related works.

| Works | Link Analysis HITS POI | Expert | Collaborative Filtering (CF) | Frequency | Variety | Repetition | Distinctiveness |
|---|---|---|---|---|---|---|---|
| TBHG [20] | ✓ | ✓ | | ✓ | | | |
| LocPref [5] | | ✓ | ✓ | ✓ | | | |
| ImpHITS [14] | ✓ | | | ✓ | ✓ | | |
| CLoRW [21] | ✓ | ✓ | | ✓ | | | |
| TAP-F [15] | ✓ | ✓ | ✓ | ✓ | | | |
| NILR [9] | ✓ | | | ✓ | | ✓ | |
| WHITS [22] | ✓ | | | ✓ | | | |
| TDCF [13] | ✓ | | ✓ | ✓ | | | |
| PPTA-RM [23] | ✓ | | ✓ | ✓ | | | |
| FCDST [27] | | | ✓ | ✓ | | | |

| Works | Link Analysis | | Collaborative Filtering (CF) | Features | | | |
| --- | --- | --- | --- | --- | --- | --- | --- |
| | HITS | | | Frequency | Variety | Repetition | Distinctiveness |
| | POI | Expert | | | | | |
| SPR [28] | | | ✓ | ✓ | | | |
| timePOI [29] | | | ✓ | ✓ | | | |
| CTF-ARA [30] | | | ✓ | ✓ | | | |
| U-CF-M [12] | | | ✓ | ✓ | | | |
| CLGRW [31] | | | ✓ | ✓ | | | |
| U-CF-MS [32] | | | ✓ | ✓ | | ✓ | |
| CULT-TF [33] | | | ✓ | ✓ | | | |
| LACF-Rec3 | ✓ | ✓ | ✓ | ✓ | ✓ | ✓ | ✓ |

## 3. Our Method

In this section, the basic notations and details of our proposed location-based recommendation system are described. As illustrated in Figure 1, our LACF-Rec3 method consists of two main phases: the offline and online phases. The first step within the offline phase involves the determination of frequency, variety, and repetition statistics related to the locations and users. Subsequently, interest scores are calculated for the locations and users using our extended HITS-based model (termed HITS-3), leveraging the statistics obtained from the preceding step. In the third step, distinctiveness and distinctive scores are identified for the users and locations.

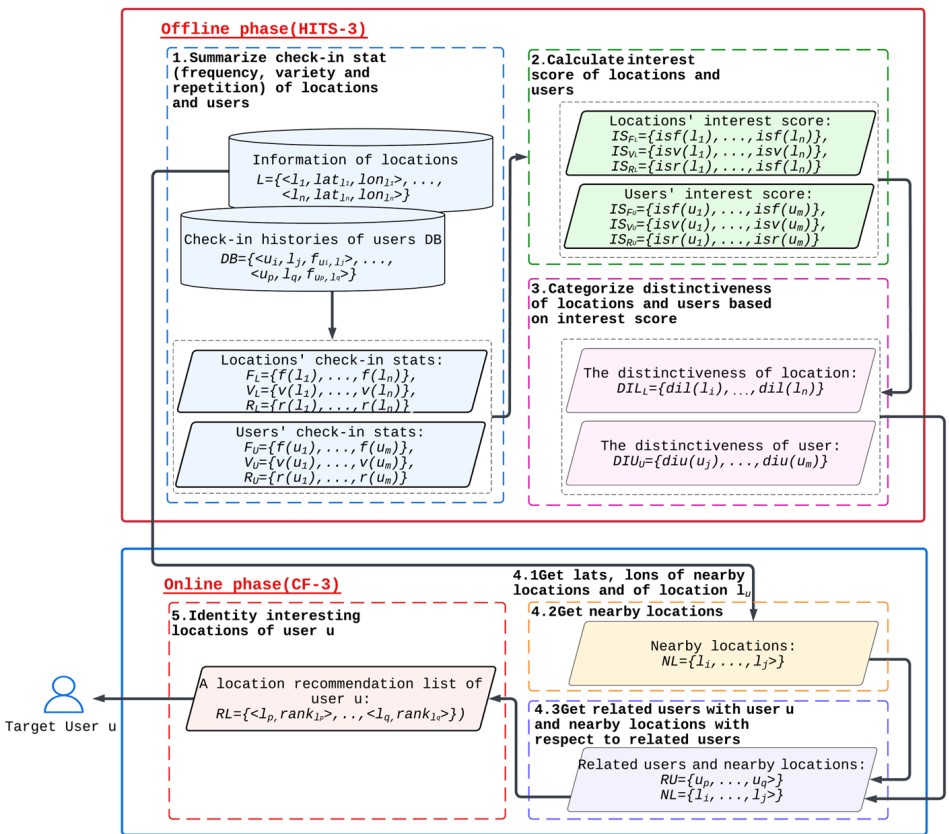

**Figure 1.** Our LACF-Rec3 method consists of two phases: offline phase with HITS-3, and online phase with CF-3.

In the online phase, upon the arrival of a target user's request for a POI recommendation list, our approach retrieves, from the database, locations proximate to the target user, u, as well as the set of users who have visited these locations at least once, RU. Next, our enhanced collaborative filtering technique (CF-3) determines the distinctiveness similarity between u and each user within RU. Finally, the locations with the highest recommendation scores are selected and ranked, constituting the top-N location list recommended for the target user.

### 3.1. Notation

Let $L\_DB = \{<l_1, lat_{l_1}, lon_{l_1}>, \ldots, <l_n, lat_{l_n}, lon_{l_n}>\}$ be a set of all $n$ locations where each location $l_j$ is associated with its latitude ($lat_{l_j}$) and longitude ($lon_{l_j}$). A set $U = \{u_1, u_2, \ldots, u_m\}$ is a set of all users (i.e., $m$ users) who checks in at least one location ($l_j$) in $L\_DB$. A record of check-in history of a user $u_i$ at a location $l_j$ is represented by a 3-tuple $<u_i, l_j, f_{u_i, l_j}>$ where $f_{u_i, l_j}$ is the visit frequency of $u_i$ at $l_j$ (i.e., the number of times that $u_i$ checks in at $l_j$). Let $DB = \{<u_1, l_1, f_{u_1, l_1}>, \ldots, <u_m, l_n, f_{u_m, l_n}>\}$ be the set of check-in histories of all users, $DB_{l_p} = \{<u_1, l_p, f_{u_1, l_p}>, \ldots, <u_m, l_p, f_{u_m, l_p}>\}$ be the set of the check-in histories of all users visiting the location $<l_p, lat_{l_p}, lon_{l_p}> \in L\_DB$, and $DB_{u_j} = \{<u_j, l_1, f_{u_j, l_1}>, \ldots, <u_j, l_n, f_{u_j, l_n}>\}$ be the set of check-in histories of a user $u_j$ visiting all $n$ locations in $L\_DB$. The current location of a target user $u$ is denoted by $<lat_u, lon_u>$.

To provide a list of interesting locations to a target user $u$ located at $<lat_u, lon_u>$, the prior collected data of $DB$, $DB_{l_p}$, $DB_{u_j}$, and $<lat_u, lon_u>$ are considered as input. Then, our LACF-Rec3 considers three visiting behaviors (i.e., the frequency, variety, and repetitions) of all users to identify interesting locations. All related notations can be described as follows.

**Definition 1 (Frequency of locations and of users).** *The checked-in frequency of $l_p$ is the total number of times that all m users in U check in at $l_p$, computed as follows.*

$$f(l_p) = \sum_{k=1}^{m} f_{u_k, l_p} \tag{1}$$

*Similarly, the check-in frequency of $u_j$ is the total number of locations that the user $u_j$ visited, computed as follows.*

$$f(u_j) = \sum_{k=1}^{n} f_{u_j, l_k} \tag{2}$$

**Definition 2 (Variety of locations and of users).** *The user variety of $l_p$ is the number of individuals ($u_k$) who check in at least once at $l_p$ retrieved from the database $DB_{l_p}$, defined as follows.*

$$v(l_p) = \sum_{k=1}^{m} va_k \text{ where } va_k = \begin{cases} 1, & \textit{if } f_{u_k, l_p} > 0 \\ 0, & \textit{otherwise} \end{cases} \tag{3}$$

*Similarly, based on the database $DB_{u_j}$, the location variety of $u_j$ is the number of locations $u_j$ visited, defined as follows.*

$$v(u_j) = \sum_{k=1}^{n} va_k \text{ where } va_k = \begin{cases} 1, & \textit{if } f_{u_j, l_k} > 0 \\ 0, & \textit{otherwise} \end{cases} \tag{4}$$

**Definition 3 (Repetitions of locations and of users).** *Based on the database $DB_{l_p}$, the checked-in repetition of $l_p$ is the number of users who have visited $l_p$ more than once, defined as follows.*

$$r(l_p) = \sum_{k=1}^{m} re_k \text{ where } re_k = \begin{cases} 1, & \textit{if } f_{u_k, l_p} > 1 \\ 0, & \textit{otherwise} \end{cases} \tag{5}$$

Similarly, based on the database $DB_{u_j}$, the number of locations $u_j$ revisited is the number of locations $u_j$ visited more than once, defined as follows.

$$r(u_j) = \sum_{k=1}^{n} re_k \text{ where } re_k = \begin{cases} 1, & \text{if } f_{u_j, l_k} > 1 \\ 0, & \text{otherwise} \end{cases} \tag{6}$$

**Definition 4 (Normalized characteristics).** *Let $F_L = \{f(l_1), f(l_2), \ldots, f(l_n)\}$ be the set of checked-in frequency of locations in L_DB. To prevent small values being ignored, and avoid the dominance of one characteristic over the other two, each checked-in frequency $f(l_p) \in F_L$ is normalized as follows*

$$nf(l_p) = \begin{cases} 0.5 + \left( \left( f(l_p)\text{-}avg \right) \times \frac{0.5}{max\text{-}avg} \right), & f(l_p) \geq avg \\ f(l_p) \times \frac{0.5}{avg}, & \text{otherwise} \end{cases} \tag{7}$$

*where avg is the average value of checked-in frequencies in $F_L$ (i.e., $avg(f(l_1), f(l_2), \ldots, f(l_n))$), and max is the maximum checked-in frequency in $F_L$ (i.e., $max(f(l_1), f(l_2), \ldots, f(l_n))$).*

*Similarly, Equation (7) is applied to normalize each check-in frequency of a user $f(u_j) \in F_U = \{f(u_1), f(u_2), \ldots, f(u_m)\}$, each user variety of a location $v(l_p) \in V_L = \{v(l_1), v(l_2), \ldots, v(l_n)\}$, each location variety of a user $v(u_j) \in V_U = \{v(u_1), v(u_2), \ldots, v(u_m)\}$, each checked-in repetition of a location $r(l_p) \in R_L = \{r(l_1), r(l_2), \ldots, r(l_n)\}$, and the number of locations $u_j$ revisited $r(u_j) \in R_U = \{r(u_1), r(u_2), \ldots, r(u_m)\}$.*

**Definition 5 (Interest score of locations and of users).** *Let $ISF_L = \{isf(l_1), isf(l_2), \ldots, isf(l_n)\}$ and $ISF_U = \{isf(u_1), isf(u_2), \ldots, isf(u_m)\}$ be the set of frequency-based interest scores of all locations and the set of all users based on their frequencies, respectively. All frequency-based interest scores in $ISF_U$ are initialized to 1. The frequency-based interest score $isf(l_p) \in ISF_L$ of $l_p \in L\_DB$ is the summation of all of the users' scores (which is the multiplication of each user's check-in frequency and prior-interest score) who have checked in at the location $l_p$. The frequency-based interest score $isf(u_q) \in ISF_U$ of $u_q \in U$ is subsequently computed by summarizing the score of locations that $u_q$ has checked in. Both values can be defined as follows.*

$$isf(l_p) = \sum_{q=1}^{m} \begin{cases} f(u_q) \times isf(u_q), & \text{if user } u_q \text{ has checked in at location } l_p \\ 0, & \text{otherwise} \end{cases} \tag{8}$$

$$isf(u_q) = \sum_{p=1}^{n} \begin{cases} f(l_p) \times isf(l_p), & \text{if user } u_q \text{ has checked in at location } l_p \\ 0, & \text{otherwise} \end{cases} \tag{9}$$

*Similarly, the variety-based and repetition-based interest score of all locations and all users can be defined as $ISV_L = \{isv(l_1), isv(l_2), \ldots, isv(l_n)\}$, $ISV_U = \{isv(u_1), isv(u_2), \ldots, isv(u_m)\}$, $ISR_L = \{isr(l_1), isr(l_2), \ldots, isr(l_n)\}$ and $ISR_U = \{isr(u_1), isr(u_2), \ldots, isr(u_m)\}$, respectively, where each interest score can be computed as follows.*

$$isv(l_p) = \sum_{q=1}^{m} \begin{cases} v(u_q) \times isv(u_q), & \text{if user } u_q \text{ has checked in at location } l_p \\ 0, & \text{otherwise} \end{cases} \tag{10}$$

$$isv(u_q) = \sum_{p=1}^{n} \begin{cases} v(l_p) \times isv(l_p), & \text{if user } u_q \text{ has checked in at location } l_p \\ 0, & \text{otherwise} \end{cases} \tag{11}$$

$$isr(l_p) = \sum_{q=1}^{m} \begin{cases} r(u_q) \times isr(u_q), & \text{if user } u_q \text{ has checked in at location } l_p \\ 0, & \text{otherwise} \end{cases} \tag{12}$$

$$isr(u_q) = \sum_{p=1}^{n} \begin{cases} r(l_p) \times isr(l_p), & \text{if user } u_q \text{ has checked in at location } l_p \\ 0, & \text{otherwise} \end{cases} \quad (13)$$

Finally, L-2 normalization is applied to $isf(l_p)$, $isf(u_q)$, $isv(l_p)$, $isv(u_q)$, $isr(l_p)$, and $isr(u_q)$.

**Definition 6 (Distinctiveness of a location and of a user).** *To point out the distinctiveness of each location $l_p \in L\_DB$, we identify the most outstanding aspect of $l_p$. The distinctiveness of $l_p$ is 1 if the frequency-based interest score is greater than the other two. The distinctiveness of $l_p$ is 2 and 3, if the variety-based and repetition-based interest score is greatest, respectively, illustrated as follows.*

$$dil(l_p) = \begin{cases} 1, & \text{if } isf(l_p) = max(isf(l_p), isv(l_p), isr(l_p)) \\ 2, & \text{if } isv(l_p) = max(isf(l_p), isv(l_p), isr(l_p)) \\ 3, & \text{if } isr(l_p) = max(isf(l_p), isv(l_p), isr(l_p)) \end{cases} \quad (14)$$

*Similarly, we identify the most outstanding aspect of $u_q$. The distinctiveness of $u_q$ is determined as follows.*

$$diu(u_q) = \begin{cases} 1, & \text{if } isf(u_q) = max(isf(u_q), isv(u_q), isr(u_q)) \\ 2, & \text{if } isv(u_q) = max(isf(u_q), isv(u_q), isr(u_q)) \\ 3, & \text{if } isr(u_q) = max(isf(u_q), isv(u_q), isr(u_q)) \end{cases} \quad (15)$$

**Definition 7 (Distinctiveness score of a user).** *Let $L_{u_q} = \{< l_i, f_{u_q,l_i}, dil(l_i) >, \ldots, < l_j, f_{u_q,l_j}, dil(l_j) >\}$ be a set of 3-tuples, containing the name of locations checked-in by $u_q \in U$, check-in frequency, and the distinctiveness of the location. Also, let $L_{u_q}^1$, $L_{u_q}^2$, and $L_{u_q}^3$ be the set of 3-tuples in $L_{u_q}$, of which distinctiveness of the location is 1, 2, and 3, respectively. The distinctiveness score of $u_q$, $ds(u_q) = < dsf(u_q), dsv(u_q), dsr(u_q) >$, indicates the distinctiveness of $u_q$ on the aspects of check-in frequency, location variety, and the number of locations $u_q$ revisited. $ds_{u_q}$ is updated as follows.*

$$dsf(u_q) = \sum_{i=1}^{|L_{u_q}^1|} f_{u_q,l_i} \quad (16)$$

$$dsv(u_q) = \sum_{j=1}^{|L_{u_q}^2|} f_{u_q,l_j} \quad (17)$$

$$dsr(u_q) = \sum_{k=1}^{|L_{u_q}^3|} f_{u_q,l_k} \quad (18)$$

**Definition 8 (Similarity between users).** *Let $u_q$ be the user who checked in at any location $l_p$ nearby the current location of the target user $u$. Thus, to identify similarity in check-in behavior between $u$ and $u_q$, the cosine similarity $sim(u, u_q)$ is determined by comparing their distinctiveness scores as follows.*

$$sim(u, u_q) = \frac{\left(dsf(u) \times dsf(u_q)\right) + \left(dsv(u) \times dsv(u_q)\right) + \left(dsr(u) \times dsr(u_q)\right)}{\sqrt{dsf(u)^2 + dsv(u)^2 + dsr(u)^2} \times \sqrt{dsf(u_q)^2 + dsv(u_q)^2 + dsr(u_q)^2}} \quad (19)$$

**Definition 9 (A recommendation score of a location based on relevant users).** *Let RU be a set of relevant users who have checked in at least once at locations nearby the current location of the target user $u$. Let SRU be a set of relevant users, where $u_j$ is the element of SRU if and only if $sim(u, u_j)$ is more than a threshold. A recommendation score of a nearby location $l_p$ is determined by considering the visiting information associated with the relevant users including (i) the similarity between $u$ and each relevant user $u_j$, $sim(u, u_j)$, (ii) the check-in frequency of $u_j$ at the location $l_p$,*

$f_{u_j,l_p}$, and (iii) the distinctiveness score of $u_j$ according to $dil(l_p)$. The recommendation score, $rs(l_p)$, of the location $l_p$ is computed as follows.

$$rs(l_p) = \sum_{j=1}^{|SRU|}(sim(u, u_j) \times f_{u_j,l_p} \times dsx(u_j)), \tag{20}$$

$$where\ dsx(u_j) = \begin{cases} dsf(u_j), & if\ dil(l_p) = 1 \\ dsv(u_j), & if\ dil(l_p) = 2 \\ dsr(u_j), & if\ dil(l_p) = 3 \end{cases}$$

**Definition 10 (A recommendation score of a location by location's interest scores).** *Let $NL = \{l_p, \ldots, l_q\}$ be a set of locations nearby the current location of the target user u. The recommendation score for the location $l_p \in NL$ is determined as the maximum interest score among the frequency-based, variety-based, and repetition-based interest scores as follows.*

$$rs(l_p) = max(isf(l_p), isv(l_p), isr(l_p)) \tag{21}$$

According to the definitions given above, we divided the system into two phases: (i) the offline phase utilizes Definitions 1–6 to determine the distinctiveness of each user and location through our HITS-3 algorithm, and (ii) the online phase applies Definitions 7–10 to create a list of recommended POIs for the target users by employing our CF-3 algorithm.

### 3.2. Offline Phase

Given the check-in database *DB* of all users' visits to various locations as input, the offline phase aims to ascertain the distinctiveness of the users and locations in terms of the frequency, variety, and repetition of check-ins. Generally, locations with a high check-in frequency correspond to popular or trendy places, of which the number of visits or revisits is high. Locations with a high variety are typically tourist attractions and landmarks that most users or tourists visit only once. Locations with numerous repeated check-ins could be associated with places users frequently visited in daily life (such as grocery stores or supermarkets) or specific venues for particular users (such as gyms, basketball courts, board game cafes, etc.). Users with a high check-in frequency are likely to be active and sociable; those with diverse check-in locations tend to seek novel experiences and avoid monotony. Finally, users with high repeated check-ins might exhibit specific preferences or strong loyalty.

In the first and second lines of Algorithm 1, the check-in statistical characteristics for each location $l_p$ and for each user $u_j$ (i.e., frequency ($f(l_p)$ and $f(u_j)$), variety ($v(l_p)$ and $v(u_j)$), and repeatedly ($r(l_p)$ and $r(u_j)$) are first calculated by Definitions 1–3 and then aggregated into the sets *LCS* and *UCS* (short for *Locations' Check-in Statistics* and *Users' Check-in Statistics*). To determine the distinctiveness of the locations and users, it is crucial to avoid the dominance of one characteristic over the others. Frequency values are usually higher than variety values, which are mostly greater than repeated check-in values. The third and fourth line of Algorithm 1 depict the normalization of each check-in statistic for each location and each user (by Definition 4), ensuring the absence of dominance. Specifically, we extended the HITS model (terms as *HITS-based on 3 check-in behaviors*, *HITS-3*) to take all three statistical characteristics (as opposed to the traditional frequency-based HITS model). This extension enables the computation of interest scores for the locations and users. As described in Definition 5, the interest score of each location $l_p$ can be expressed as (i) the frequency-based interest score, $isf(l_p)$, (ii) variety-based interest score, $isv(l_p)$, and (iii) interest score based on repeated check-ins, $isr(l_p)$, which forms a 3-tuple $< isf(l_p),\ isv(l_p),\ isr(l_p) >$. Similarly, each user $u_i$ also has three interest scores in a 3-tuple form, $< isf(u_i),\ isv(u_i),\ isr(u_i) >$. With the HITS-based model, these interest scores for the locations and users are iteratively computed. As the iteration progresses, the value of each interest score increases. As in the fifth to eighth line, we applied L2-normalization

to all of the interest scores to counteract exponential growth and allow for quicker score convergence. In the last step, in the sixth and seventh line of Algorithm 1, the highest interest score of each location and each user is determined and utilized to identify the distinctiveness of both the locations and the users. If the frequency-based interest score for the location $l_p$ is the highest, it stands out for the check-in frequencies (i.e., $dil(l_p)$ is "1"). Similarly, $dil(l_p)$ is assigned "2" if the location's variety-based interest score is the highest, and "3" if its repetition-based score is the highest. The determination of $diu(u_q)$ follows a similar procedure. The complete algorithm for the offline phase is outlined in Algorithm 1.

---

**Algorithm 1.** Offline phase Algo

---

**Input:**
    A check-in database $DB = \{< u_i, l_j, f_{u_i,l_j} >, \ldots, < u_p, l_q, f_{u_p,l_q} >\}$
    A location database $L\_DB = \{< l_1, lat_{l_1}, lon_{l_1} >, \ldots, < l_n, lat_{l_n}, lon_{l_n} >\}$
**Output:**
    Interest scores of locations and users $IS_{F_L}, IS_{F_U}, IS_{V_L}, IS_{V_U}, IS_{R_L}, IS_{R_U}$
    A distinctiveness of each location $dil(l_p)$ and user $diu(u_q)$
1: compute location's check-in stats $LCS = \{< f(l_1), v(l_1), r(l_1) >, \ldots, < f(l_n), v(l_n), r(l_n) >\}$ (by Definitions 1–3)
2: compute user's check-in stats $UCS = \{< f(u_1), v(u_1), r(u_1) >, \ldots, < f(u_m), v(u_m), r(u_m) >\}$ (by Definitions 1–3)
3: normalize $LCS$ (by Definition 4)
4: normalize $UCS$ (by Definition 4)
5: **for** each $t$ time **do**
6: compute interest score of locations and interest score of users, $IS_{F_L}$,
7: $IS_{F_U}, IS_{V_L}, IS_{V_U}, IS_{R_L}, IS_{R_U}$ (by Definition 5)
8: **end for**
9: identify distinctiveness $dil(l_p)$ of each location in $L\_DB$ (by Definition 6)
10: identify distinctiveness $diu(u_q)$ of each user in $U$ (by Definition 6)

---

As shown in Figure 2, the check-in history from a prior collected check-in database contains three users ($u_1$, $u_2$ and $u_3$) and three locations ($l_1$, $l_2$ and $l_3$). User $u_1$ checks in at $l_1$ and $l_3$ four and two times, respectively. User $u_2$ checks in at $l_1$, $l_2$, and $l_3$ one, five, and four times, respectively. User $u_3$ checks in once at $l_1$. In step 1, the visiting frequencies ($F_L$) of $l_1$, $l_2$, and $l_3$ are calculated as $4 + 1 + 1 = 6$, $0 + 5 + 0 = 5$, and $4 + 2 + 0 = 6$. The user varieties ($V_L$) of $l_1$, $l_2$, and $l_3$ are then computed as $1 + 1 + 1 = 3$, $0 + 1 + 0 = 1$, and $1 + 1 + 0 = 2$. Next, the number of repeated check-ins ($R_L$) of $l_1$, $l_2$, and $l_3$, or the number of users that visit the locations more than once, are computed as $1 + 0 + 0 = 1$, $0 + 1 + 0 = 1$, and $1 + 1 + 0 = 2$, respectively. $F_L$, $V_L$, and $R_L$ are stored in $LCS$. Similarly, the check-in frequencies ($F_U$) of $u_1$, $u_2$, and $u_3$ are computed as $4 + 0 + 2 = 6$, $1 + 5 + 4 = 10$, and $1 + 0 + 0 = 1$. The location varieties ($V_U$) of $u_1$, $u_2$, and $u_3$ are then computed as $1 + 0 + 1 = 2$, $1 + 1 + 1 = 3$, and $1 + 0 + 0 = 1$. Next, the number of revisits ($R_U$) of $u_1$, $u_2$ and $u_3$ are counted as $1 + 0 + 1 = 2$, $0 + 1 + 1 = 2$, and $0 + 0 + 0 = 0$, respectively. $F_U$, $V_U$, and $R_U$ are stored in $UCS$ as shown in the first and second lines of Algorithm 1. In the second step, the visiting frequencies $l_1$, $l_2$, and $l_3$ are normalized as $0.5 + \left((6 - 5.67) \times \frac{0.5}{6 - 5.67}\right) = 1$, $5 \times \frac{0.5}{5.67} = 0.4$, and $0.5 + \left((6 - 5.67) \times \frac{0.5}{6 - 5.67}\right) = 1$, where *avg* is 5.67 and *max* is 6. Similarly, the user varieties of $l_1$, $l_2$, and $l_3$ are $0.5 + \left((3 - 2) \times \frac{0.5}{3 - 2}\right) = 1$, $1 \times \frac{0.5}{2} = 0.5$, and $0.5 + \left((2 - 2) \times \frac{0.5}{3 - 2}\right) = 0.5$, where *avg* is 2 and *max* is 3. Next, the numbers of repeated check-ins of $l_1$, $l_2$, and $l_3$ are computed as $1 \times \frac{0.5}{1.3} = 0.38$, $1 \times \frac{0.5}{1.3} = 0.38$, and $0.5 + \left((2 - 1.3) \times \frac{0.5}{2 - 1.3}\right) = 1$, where *avg* is 1.3 and *max* is 2, respectively. The check-in frequencies of $u_1$, $u_2$, and $u_3$ are normalized as $0.5 + \left((6 - 5.67) \times \frac{0.5}{10 - 5.67}\right) = 0.54$, $0.5 + \left((10 - 5.67) \times \frac{0.5}{10 - 5.67}\right) = 1$, and $1 \times \frac{0.5}{5.67} = 0.09$, where *avg* is 5.67 and *max* is 10. The location varieties of $u_1$, $u_2$, and $u_3$ are normalized as $0.5 + \left((2 - 2) \times \frac{0.5}{3 - 2}\right) = 0.5$, $0.5 + \left((3 - 2) \times \frac{0.5}{3 - 2}\right) = 1$, and $1 \times \frac{0.5}{2} = 0.25$, where *avg* is 2 and *max* is 3. Finally, the number of revisits of $u_1$, $u_2$, and $u_3$ are normalized as $0.5 + \left((2 - 2) \times \frac{0.5}{2 - 2}\right) = 1$, $0.5 + \left((2 - 2) \times \frac{0.5}{2 - 2}\right) = 1$, and $0$, where *avg* is 2 and *max* is 2,

respectively, as shown in the third and fourth lines of Algorithm 1. Next, in the third step, the interest score of the locations and users are computed using our HITS-3 model, as described in the fifth to eighth lines of Algorithm 1. Three interest scores of each location are specified in a 3-tuple form (i.e., $< isf(l_p), isv(l_p), isr(l_p) >$). The interest scores of $l_1$, $l_2$, and $l_3$ are computed as <0.3, 0.5, 0.2>, <0.4, 0.3, 0.1>, and <0.2, 0.5 and 0.7>, respectively. The distinctiveness of each location ($dil(l_p)$) is determined according to the ninth line of Algorithm 1. Therefore, the distinctiveness of $l_1$, $l_2$, and $l_3$ are in the variety, frequency, and repetition aspects, respectively. Similarly, three interest scores of each user are also specified in a 3-tuple form (i.e., $< isf(u_i), isv(u_i), isr(u_i) >$). The interest scores of $u_1$, $u_2$, and $u_3$ are <0.3, 0.2, 0.4>, <0.6, 0.5, 0.4>, and <0.2, 0.1, 0.1>, respectively. The distinctiveness of each user ($diu(u_q)$) is determined according to the tenth line of Algorithm 1. The distinctiveness of $u_1$, $u_2$, and $u_3$ are in the repetition, frequency, and frequency aspects, respectively.

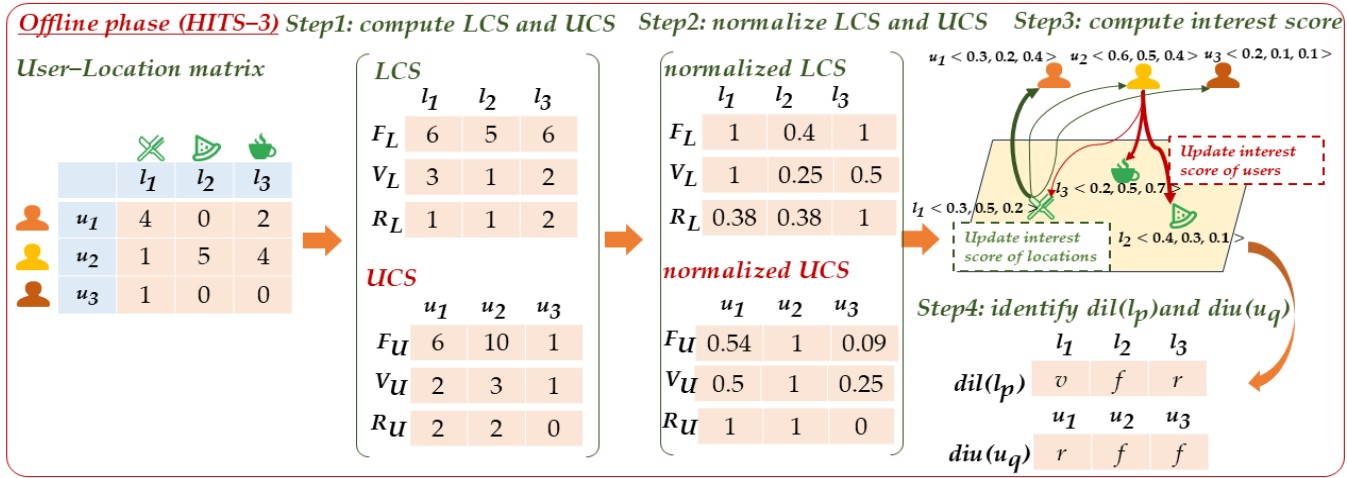

**Figure 2.** Example of the offline phase in the LACF-Rec3.

### 3.3. Online Phase

To recommend locations to a target user $u$, the user's current location, $< lat_u, lon_u >$, is required along with the request. Subsequently, nearby locations within the distance of $dt$ from $u$ (the maximum distance specified by $u$), $NL = \{l_p, \ldots, l_q\}$, are initially identified (as shown in the first line of Algorithm 2). The users who have checked in at least once at a nearby location $l_p \in NL$ (also referred to as relevant users), $RU = \{u_i, \ldots, u_j\}$, are then retrieved (as shown in the second line of Algorithm 2). Next, the three distinctiveness scores of each user $u_q \in RU$, $ds(u_q)$, are computed on the aspects of check-in frequency, location variety, and the number of locations the user revisits, respectively (in accordance with Definition 7). The frequency-based distinctiveness, $dsf(u_q)$, of user $u_q$ is determined by summing the visit frequency of $u_q$ at $l_j$, which stands out in the aspect of checked-in frequency (i.e., $dil(l_j) = 1$). Similarly, the variety-based distinctiveness score, $dsv(u_q)$, of user $u_q$ is calculated by summing the visit frequency of $u_q$ at $l_j$, which stands out in the aspect of location variety (i.e., $dil(l_j) = 2$). Finally, the repetition-based distinctiveness score, $dsr(u_q)$, of user $u_q$ is derived by summing the visit frequency of $u_q$ at $l_j$, which stands out in the aspect of the number of revisits (i.e., $dil(l_j) = 3$).

Subsequently, the similarity, $sim(u, u_q)$, between the distinctiveness of the target user $u$ and each user $u_q \in RU$ is calculated according to Definition 8. A user $u_q$ is characterized as an "irrelevant user with respect to the target user $u$" and is not selected from $RU$ to be in $SRU$ if its similarity to $u$ falls below the predefined similarity threshold $\delta$ (as stated in the third line of Algorithm 2). Note that the target user $u$ will be considered as a cold-start user, if they lack a check-in history or if there is no "relevant user with respect to the target user $u$" (i.e., $SRU = \varnothing$).

---

**Algorithm 2.** Online pvhase Algo

---

**Input:**

      A check-in database $DB = \{< u_i, l_j, f_{u_i, l_j} >, \ldots, < u_p, l_q, f_{u_p, l_q} >\}$

      A location database $L\_DB = \{< l_1, lat_{l_1}, lon_{l_1} >, \ldots, < l_n, lat_{l_n}, lon_{l_n} >\}$

      Interest scores of location ns and users $IS_{F_L}, IS_{F_U}, IS_{V_L}, IS_{V_U}, IS_{R_L}, IS_{R_U}$

      A distinctiveness of each location $dil(l_p)$ and user $diu(u_q)$

      A number of recommendations $N$

      A target user $u$ with her current location $< lat_u, lon_u >$

      A distance condition on recommended location, $d$

      A similarity threshold, $\delta$

**Output:**

      A recommended list $RL = \{< l_1, rs(l_1) >, < l_2, rs(l_2) >, \ldots, < l_n, rs(l_n) >\}$

1: identify $NL = \{l_p, \ldots, l_q\}$, the set of locations nearby the current location $< lat_u, lon_u >$ of $u$ (within distance $d$)

2: identify $RU = \{u_i, \ldots, u_j\}$, the set of relevant users visiting the location in $NL$ at least once

3: determine the distinctiveness score of each user, $u_i$, in $RU$ and the similarity of $u_i$ and $u$. Then, identify $SRU$ by selecting the users from $RU$ to be in $SRU$ based on their distinctiveness with the similarity threshold $\delta$ (by Definitions 7 and 8)

4: compute $RS_{NL} = \{rs(l_p), \ldots, rs(l_q)\}$ the set of recommendation scores of all locations in $NL$ (if $RU \neq \varnothing$ applies Definition 9, otherwise, applies Definition 10)

5: determine $RL = \{< l_p, rank_{l_p} >, \ldots, < l_q, rank_{l_q} >\}$, the set of $N$ nearby locations with the associated ranks

6: recommend $RL$ to the target user $u$

---

Stated in the fourth line of Algorithm 2, the recommended locations for the target user, $u$, can be determined in two cases:

(Case 1) In the case that $u$ is associated with relevant users ($SRU \neq \varnothing$), we extended collaborative filtering (named CF-3) to address the challenge of sparsity. Instead of the similarity in visited locations, CF-3 determines the resemblance between $u$ and each $u_q \in SRU$ by considering the similarity of the distinctiveness of location $l$ ($dil(l)$). CF-3 takes into account the distinctive scores of relevant users $u_q$ ($dsx(u_q)$) in order to recommend intriguing locations. This approach is based on the assumption that $u$ might share similar preferences and tends to visit at the same locations as their relevant users. For each checked-in location $l_j$, its recommendation score is calculated by considering the similarity between $u$ and each user $u_q$ (who check-in at $l_j$), the check-in frequency of user $u_q$ at location $l_j$, and the distinctiveness score of $u_q$ on the aspect of the distinctiveness $di(l_j)$ of location $l_j$ (by defined in Definition 9).

(Case 2) In the case where $u$ has no relevant users (indicating a cold-start scenario), our method relies on the interest score associated with each location $l_j \in L\_DB$ (i.e., $< isf(l_j), isv(l_j), isr(l_j) >$ (provided by Algorithm 1)). Subsequently, the recommendation score of location $l_j$ is determined by selecting the highest score among the three interest scores (as specified by Definition 10).

After considering all locations (whether in Case 1 or Case 2), only top-$N$ locations with the highest recommendation scores are selected to be included in the recommendation list for the target user $u$, as outlined in the fifth line of Algorithm 2.

As depicted in Figure 3, when a target user $u$ requests a list of recommended POIs, the system first retrieves the set of locations nearby her current location ($NL$) (i.e., $l_1$, $l_2$, and $l_3$, and the set of relevant users ($RU$) (i.e., $u_1$, $u_2$, and $u_3$). This process is described in the first and second lines of Algorithm 2. In the second step, the similarities of $u$ and each user in $RU$ are calculated based on the distinctiveness score. The similarity value of $u$ and $u_1$ ($sim(u, u_1)$) is $\frac{(0 \times 0)(2 \times 4)(0 \times 2)}{\sqrt{0^2 + 2^2 + 0^2} \times \sqrt{0^2 + 4^2 + 2^2}} = 0.89$, $sim(u, u_2)$ is $\frac{(0 \times 5)(2 \times 1)(0 \times 4)}{\sqrt{0^2 + 2^2 + 0^2} \times \sqrt{5^2 + 1^2 + 4^2}} = 0.15$, and $sim(u, u_3)$ is $\frac{(0 \times 0)(2 \times 1)(0 \times 2)}{\sqrt{0^2 + 2^2 + 0^2} \times \sqrt{0^2 + 1^2 + 2^2}} = 0.44$. In the third step, if the similarity threshold $\delta$ is set to 0.4, $u_2$ is not selected to be in $SRU$, as outlined in the third line of Algorithm 2. Next, in the fourth step, the recommendation scores ($RS_{NL}$) of $NL$ are computed. As a

result, the recommendation score of $l_2$ and $l_3$ are $(0.89 \times 5 \times 0.6) + (0.44 \times 0 \times 0.2) = 2.67$, and $(0.89 \times 2 \times 0.6) + (0.44 \times 0 \times 0.2) = 1.07$, respectively. In the final step, the set of *N* nearby locations (*RL*) is ranked, with $l_3$ holding the top rank and $l_2$ taking the second to be recommended to the target user *u*.

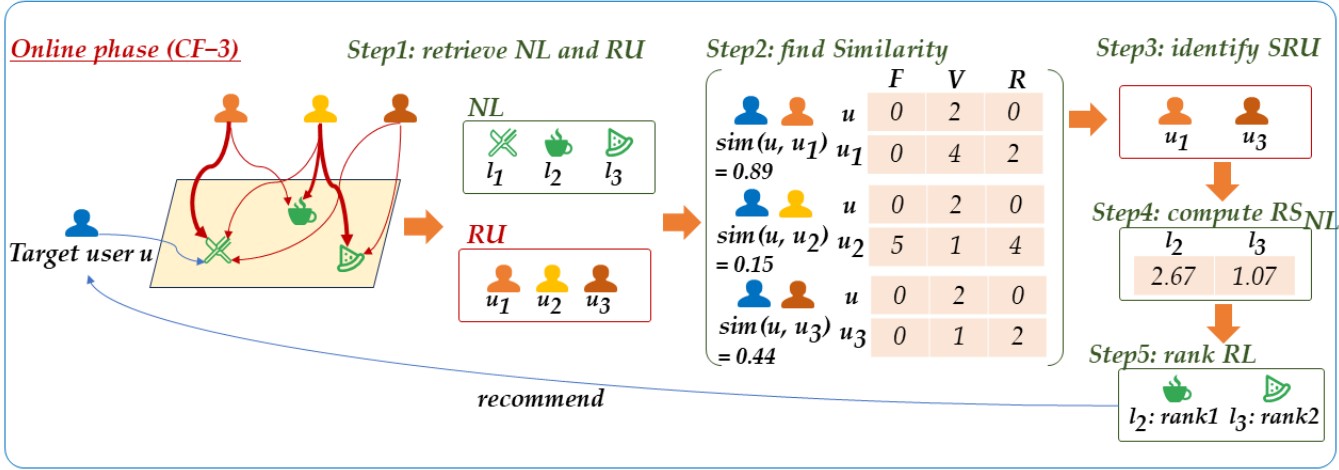

**Figure 3.** Example of the online phase in LACF-Rec3.

## 4. Experiments

In this section, we describe the experimental setup and evaluation methodologies. The details of the dataset used for these experiments are explained. We conducted a comparative analysis between our method and four baseline approaches including user-based collaborative filtering for POI recommendation (UCF), personalized POI recommendation using memory-based preferences and POI stickiness (U-CF-MS), hypertext-induced topic-search-based recommendation (HITS), and tensor decomposition-based collaborative filtering (TDCF). The evaluation method using the four metrics (i.e., precision, recall, NDCG, and matching ratio) is also explained. Finally, we report the experimental results and discuss the effectiveness of our proposed method.

### 4.1. Experiment Setup and Evaluation Methodologies

#### 4.1.1. Dataset

To evaluate the performance of our proposed method, we employed a real-world dataset obtained from Foursquare, containing historical check-in records in Tokyo spanning from April 2012 to February 2013 (https://sites.google.com/site/yangdingqi/home/foursquare-dataset accessed on 1 August 2022) [34,35]. Within the dataset, a range of check-in activities were recorded across diverse categories including food establishments (e.g., restaurants, cafes, bars), transportation facilities (e.g., bus, train, and subway stations, airports), nightlife venues (e.g., nightclubs), and more. However, it was observed that a significant number of check-ins occurred at transportation sites, primarily associated with daily routines such as commuting to work or school, rather than places visited in special occasions or by users looking for new experiences. Based on this observation, we opted to utilize only check-in data associated with food-related locations. This choice allowed us to focus on the recommendation of novel experiences to users.

Each record of the check-in data pertains to a user who has checked in at a location and contains six fields: user identifier, location name, location category, latitude of location, longitude of location, and check-in timestamp. Our method is based on applying the users' expertise in certain geographical zones to create tailored recommendation lists for the target users. We thus categorized the locations into five areas (i.e., *Chiyoda*, *Shibuya*, *Minato*, *Shinjuku*, and *Kawasaki*) by using *Google Maps Geocoding API* [36]. These areas were subsequently incorporated into the users' check-in database.

Table 2 presents the information regarding the check-in activities of 2169 users at 11,071 food-related venues in Tokyo. Some users may have check-in activities at various cities. The table also shows the analysis of the check-in histories associated with each area. *Chiyoda* was the area with the highest values in terms of the check- in users, checked-in locations, total check-ins, average check-ins per location, and average check-ins per user. Those of *Shibuya*, *Minato*, and *Shinjuku* were closely aligned, and relatively smaller than those of *Chiyoda*. Finally, *Kawasaki* had the lowest values across these aspects.

**Table 2.** The characteristics of the dataset.

| Location | Users (Existing Users/ Cold-Start Users) | Locations | Total Check-Ins | Check-In per Locations | Check-In per Users |
|---|---|---|---|---|---|
| Chiyoda | 1702 (211/27) | 2826 | 15,029 | 5.32 | 8.83 |
| Shinjuku | 1352(136/36) | 2348 | 9051 | 3.85 | 6.69 |
| Minato | 1264 (115/27) | 2556 | 8695 | 3.40 | 6.89 |
| Shibuya | 1240 (127/27) | 2222 | 8413 | 3.79 | 6.78 |
| Kawasaki | 537 (52/9) | 1119 | 4259 | 3.81 | 7.93 |
| **Total** | **2169 (356/40)** | **11,071** | **45,447** | **4.11** | **20.95** |

In our experiments, we employed a 5-fold cross-validation technique on the dataset, wherein 20% of the check-in histories were randomly selected as the testing dataset while the remaining 80% served as the training dataset.

### 4.1.2. Baseline Methods

As our method is a hybrid of link analysis and collaborative filtering techniques, we chose to compare it with the UCF [29] and U-CF-MS [32] methods because they are memory-based collaborative filtering techniques. UCF is a traditional and widely-employed collaborative filtering technique. U-CF-MS is a recently proposed user-based collaborative filtering approach geared toward individual POI recommendations, and it utilizes the number of repetitions, aligning with our proposed methodology.

Additionally, two more methods, HITS and TDCF, were also included in our comparative analysis. The HITS-based model [18] stands as the traditional and renowned algorithm developed for the task of discovering quality POI, based on a link analysis technique. Finally, TDCF [13] recommends the list of POIs using the most recently proposed hybrid method incorporating collaborative filtering and link analysis methods like our hybrid strategy. The overview of each method is described below:

*User-based collaborative filtering for POI recommendation (UCF)* [29]: This user-based collaborative filtering approach initially creates a user–location matrix, where each entry denotes the probability of a particular user visiting a particular POI. Recommendations are then generated by considering the similarity between the users' check-in histories.

*Personalized POI recommendation using memory-based preferences and POI stickiness (U-CF-MS)* [32]: This user-based collaborative filtering approach considers the recent check-in POIs of the users (memory-based) and the revisit behavior (repeated check-ins) of each POI (POI stickiness) to generate recommendations.

*Hypertext-induced-topic-search-based recommendation (HITS)* [18]: This link analysis-based approach initially constructs a user–POI network to determine the interest scores of POIs and subsequently recommend the top-N most interesting POIs within a geospatial distance.

*Tensor decomposition based collaborative filtering (TDCF)* [13]: This approach combines link analysis and collaborative filtering approaches considering the user check-in histories including user-id (username), location categories that the user checked in, and check-in time (called time slot). Additionally, its link-analysis component leverages the location popularity and distances between locations.

*Our hybrid method (LACF-Rec3)*, which combines link analysis and collaborative filtering, considers three visiting behaviors to recommend locations for both existing and cold-start users. For existing users, we introduced a novel collaborative filtering technique (CF-3) that utilizes distinctiveness similarity to recommend POIs personalized to the target user, although they might not be popular among tourists. For cold-start users, an extended version of the HITS-based model (HITS-3) was proposed. This enhanced model incorporates all three visiting behaviors—frequency, variety, and repetition—to generate the top-ranked POIs. These recommended venues are not only characterized by a large number of visits but also by a broad range of individual visitors and a significant number of revisits.

### 4.1.3. Evaluation Methods

The performance of POI recommendation is examined through the assessment of three aspects: (i) accuracy, (ii) ranking recommendation, and (iii) matching of the recommended POIs with target users retrieved from the testing dataset.

Since the dataset does not include the current location of the target user, we employed the minimum bounding rectangle (MBR) technique [5,15] to create such information. The technique involves identifying a boundary of locations in the testing dataset, shown as the dash line (derived from location A and B) in Figure 4. The target user's current location is denoted by the green circle. All POIs within the boundary—whether depicted in red or black—can be considered as candidate POIs for recommendation purposes. The red POIs represent places where the target user from the testing dataset has previously checked in, referred to as ground truth POIs. The black POIs represent ordinary POIs located within the area but not visited by the target user. The red POIs with the green outline (termed recommended ground truth POIs) are the venues recommended to the target user, who has already visited them at least one. Similarly, the black ones outlined in green are the places recommended to the target user, although they have never visited the places. For instance, in Figure 4, there are a total of seven ground truth POIs. Assuming that the target user $u$ is positioned within the green circle and requests a POI recommendation list, a recommender system recommends a list of the top five recommended POIs, consisting of three ground truth POIs and two ordinary POIs: three red and two black POIs with the green outline.

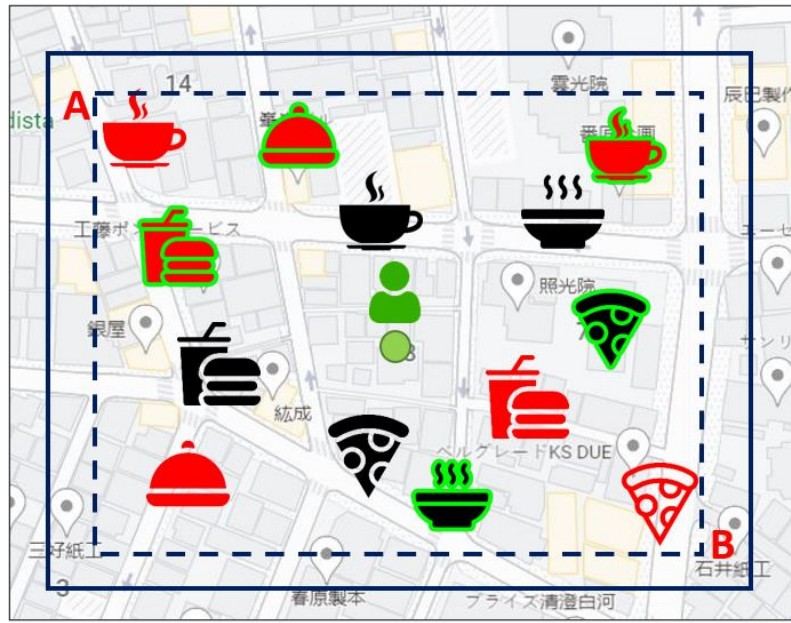

**Figure 4.** Recommendation effectiveness evaluation method using MBR.

Two common criteria (i.e., the precision and recall) (as in Equations (22) and (23)), [27,32] are adopted to evaluate the accuracy of recommendation. The precision is the ratio of

those recommended on the ground truth POIs to the total number of recommended items (denoted as *N*). The recall is the ratio of those recommended on the ground truth POIs to the total number of ground truth POIs.

$$Precision = \frac{number\ of\ Recommended\ Ground\ Truths}{number\ of\ Recommended\ POIs} \tag{22}$$

$$Recall = \frac{number\ of\ Recommended\ Ground\ Truths}{number\ of\ Ground\ Truths} \tag{23}$$

NDCG (normalized discounted cumulative gain) [16] is also adopted as a metric that evaluates the quality of the ranking of the recommended locations. Discounted cumulative gain (DCG) measures of ranking quality consider both the relevance of the recommended locations and their positions in the ranking. It is calculated by summing the relevance of each recommended location at each position in the ranking, but discounting the contributions of the locations that appear lower in the list. The ideal discounted cumulative gain (IDCG) represents the best possible DCG score that can be achieved for a given set of recommended locations. The NDCG is determined by the normalized DCG by IDCG, thus ranging from 0 to 1. The NDCG is formally given by Equations (24)–(26):

$$NDCG = \frac{DCG}{IDCG} \tag{24}$$

$$DCG = \sum\nolimits_{i=1}^{n} \frac{2^{rel_i} - 1}{log_2(i+1)} \tag{25}$$

$$IDCG = \sum\nolimits_{i=1}^{|POI_{REL}|} \frac{2^{rel_i} - 1}{log_2(i+1)} \tag{26}$$

where *N* denotes the number of POIs to be recommended, $rel_i$ represents the relevancy of the *i*th POI in the recommended list, and $POI_{REL}$ signifies the ideal list of POIs containing the top-N relevant POIs among all of the considered POIs.

For the last metric, the quality of the recommendation list is evaluated by the proposed matching ratio metric. Precision and recall, while commonly used, may not be able to holistically reflect the performance of recommendation systems within the context of the simulated experiments. This is due to the nature of the Foursquare data that were collected from users without the influence of any recommendation system. Consequently, the precision and recall calculations were based solely on the comparison of the recommended list with the list of locations where the target users have checked in, without actual user interaction with the recommendation system. To summarize, these two metrics may not entirely capture the system's performance under some circumstances, particularly when the system could suggest places that align with the preferences of the target users, but such places are not included as ground truth POIs in the Foursquare dataset. The reason behind their absence might be that (i) the target users might have visited these places but forgot to record the check-in on the platform, and (ii) the places were not known or had never been recommended to the target users for consideration.

Recognizing these limitations, we proposed the "matching ratio" as a metric to measure the extent to which POI recommendation lists align with the user preferences. It is determined by determining the ratio of (i) the number of POIs whose distinctiveness matches with the target user's distinctiveness to (ii) the number of recommended POIs, as shown in Equation (27). The ratio value is in the range of 0–100%: higher values indicate better matching between the recommended POIs and the users' preferences.

$$Matching\ ratio = \frac{number\ of\ POIs\ with\ matched\ distinctiveness}{number\ of\ Recommended\ POIs} \times 100 \tag{27}$$

## 4.2. Experimental Results and Discussion

In this part, we examine the performance of our LACF-Rec3 method by assessing the top-N recommended locations, where N ranges from 5 to 20, across six areas: Tokyo, Chiyoda, Shinjuku, Minato, Shibuya, and Kawasaki. We conducted a comparative analysis against the selected baseline methods.

### 4.2.1. Recommendation for Cold-Start Users

We started our examination by focusing on the recommendations for cold-start users. This group of users presents a distinct challenge because of the lack of a check-in history or relevant users. As a result, we have no advantage of leveraging user profiles to create personalized recommendations. In response to this challenge, we determined high-quality POIs in proximity to the user's current location by using the distinctiveness of the POI across multiple aspects, instead of the check-in history or relevant users. As shown in Figures 5–8, our proposed method has mostly higher precision and recall than HITS by 63.30% and 59.21%, respectively.

Previous studies have introduced POI recommendation algorithms that primarily focus on check-in frequency, resulting in recommended lists influenced by the popularity of locations. However, it is worth noting that users often possess diverse preferences beyond mere popularity. Certain individuals may be interested in other aspects such as the places that local experts often visit. In an ideal scenario, an optimal recommendation would include perfect destinations that excel across all aspects (i.e., being high popularly, visited by diverse individuals, and consistently revisited). However, most places could be distinctive only in specific aspects. For example, some locales might be highly frequently visited by numerous individuals, albeit a small number of revisits; thus, standing out in only two aspects. Others could attract visitors daily. Although its user variety is limited, it has high frequency rate and maintains a consistent base of returning visitors. Similarly, certain places could be a less frequently visited destination, but its visitor groups are diverse and has a loyal base of returning visitors. Our proposed method accommodates these various aspects of locations to provide recommendations that meet these multifaceted user preferences.

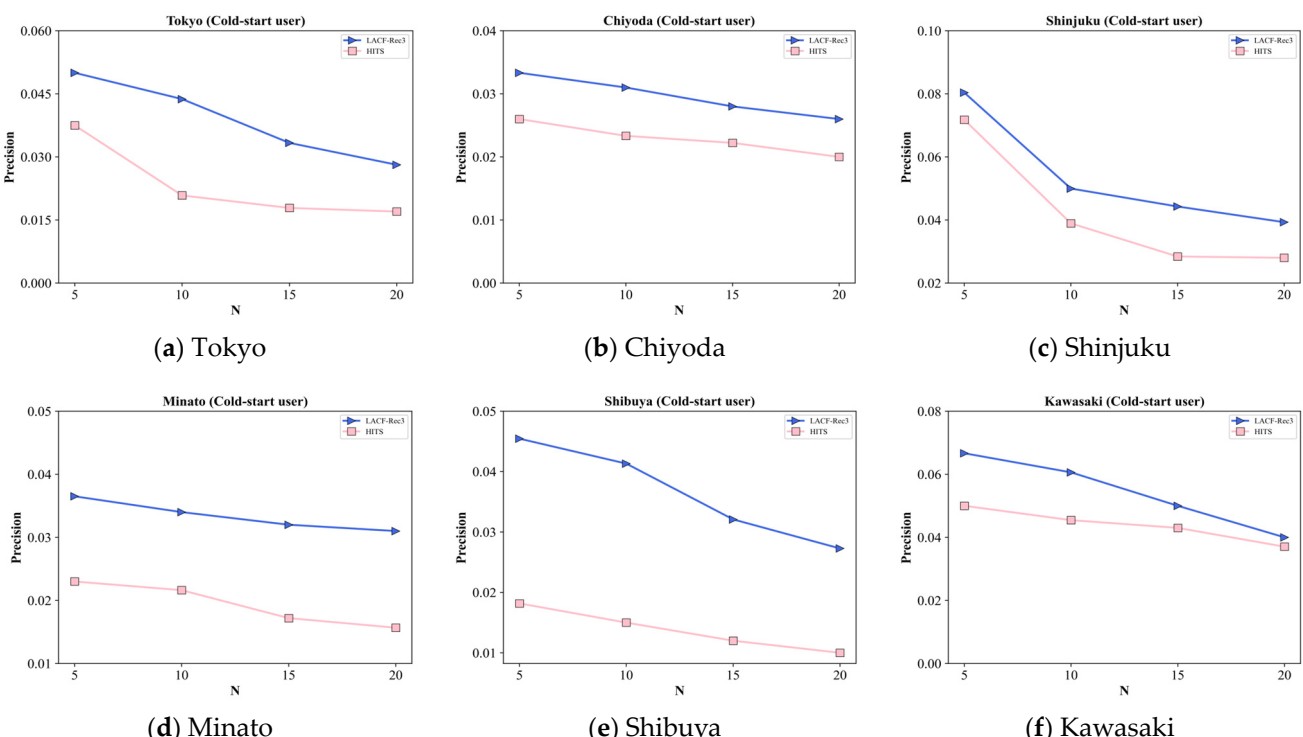

**Figure 5.** Precision metric for cold-start users with respect to the recommendation numbers.

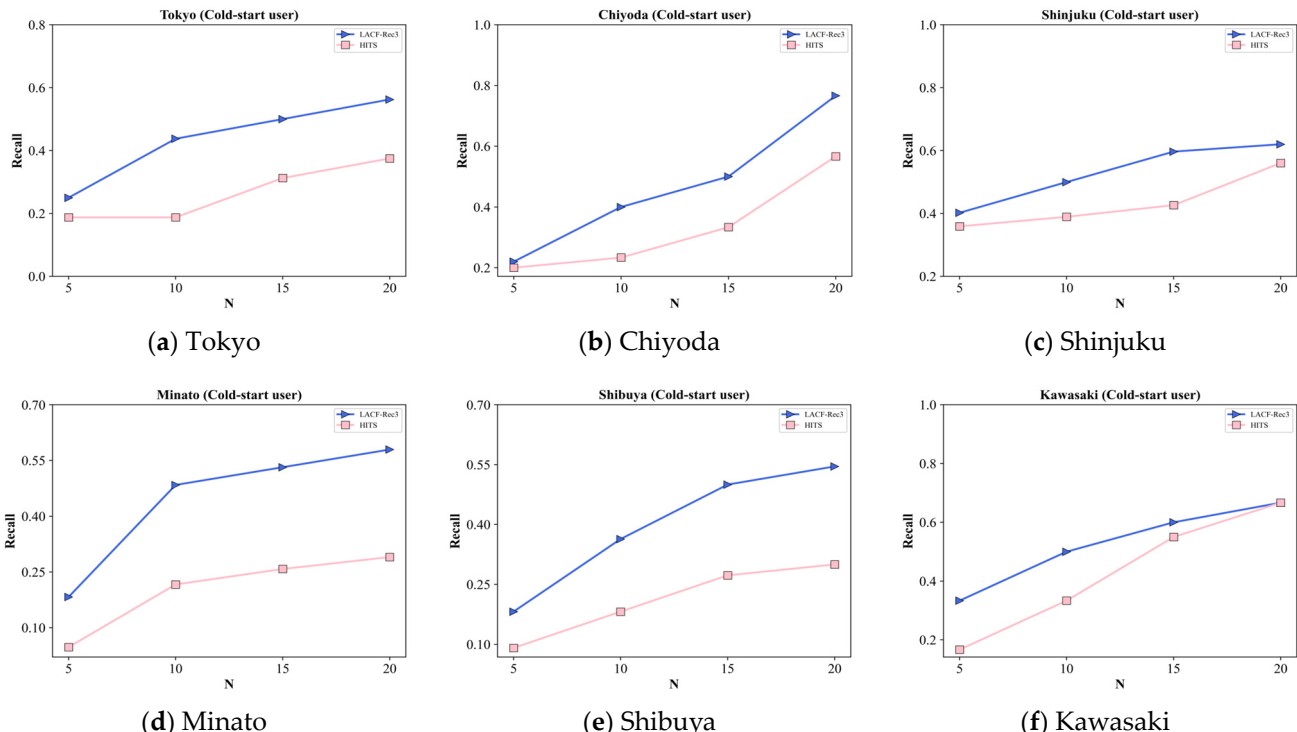

**Figure 6.** Recall metric for cold-start users with respect to the recommendation numbers.

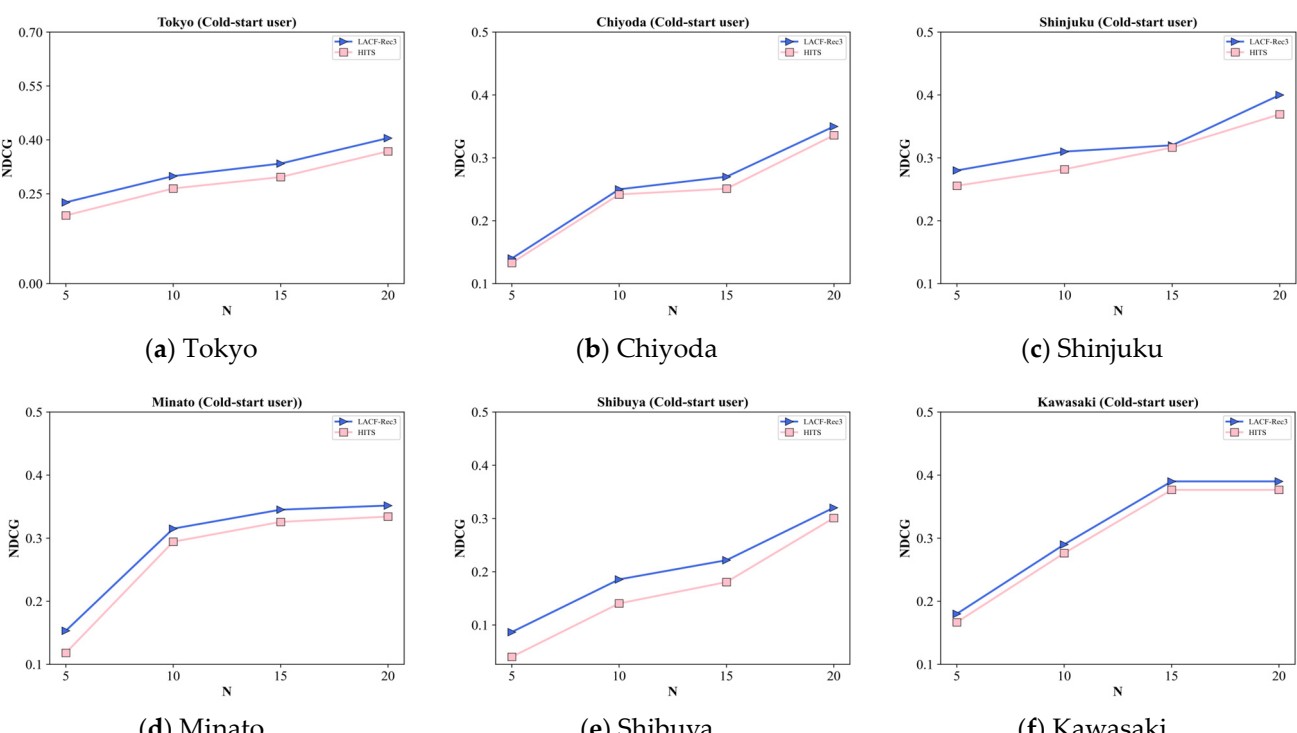

**Figure 7.** NDCG metric for cold-start users with respect to the recommendation numbers.

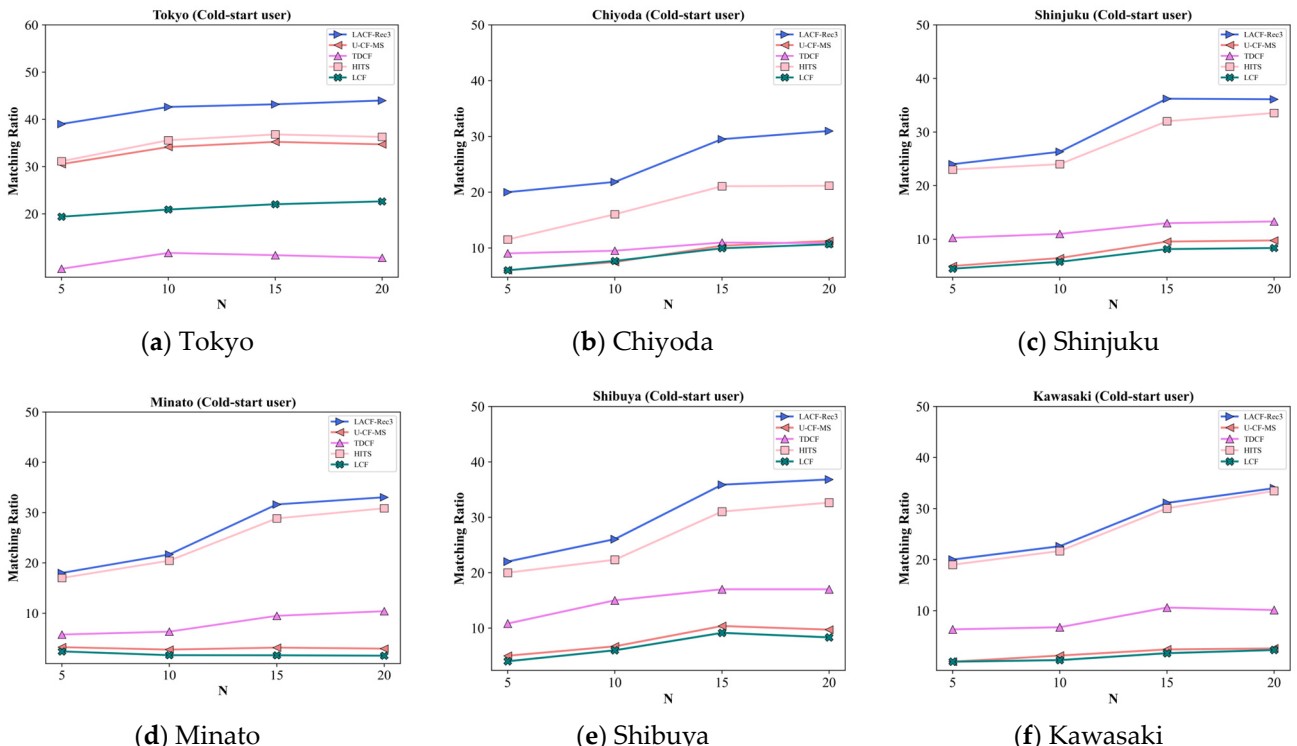

**Figure 8.** Matching ratio metric for cold-start users with respect to the recommendation numbers.

In addition, the ranks of the location recommendations selected by the user is higher than that of the previous method. The ranking accuracy evaluation using NDCG shows that the proposed algorithm was able to rank better than the HITS based model with 8.88%.

To evaluate the extent to which the location recommendations matched users' preferences, these were based on the matching ratio metric. According to the experimental results, our proposed method demonstrates the capability to recommend POIs that closely align with the user's preferences, particularly within the range of 25% to 50%. It outperforms U-CF-MS, TDCF, HITS, and UCF by 521.51%, 193.78%, 17.82%, and 793.39%, respectively.

Note that in the best case, our proposed method was able to provide recommendation lists that matched the users' preferences by 50%, meaning more than half of the recommended POIs are distinctive in the aspect different from those of the target users. This limitation stems from the fact that LACF-Rec3 determines the most distinctive aspect, which is then utilized by HIT-3, as opposed to simultaneously considering all three aspects. Our future work will focus on addressing this issue to bridge the gap.

In addition, we evaluated and compared our method with HITS as opposed to U-CF-MS, TDCF, and LCF. This choice stemmed from the inappropriateness of U-CF-MS and LCF for recommending locations for cold-start users, given their reliance on user profiles. Although TDCF is an extended version of HITS, it also requires a user profile to identify the active area of the target user. Hence, it cannot generate a POI recommendation list for cold-start users. The experimental results show that our method outperformed HITS in all of these areas. Especially in areas where cold-start users are prevalent such as Minato and Shibuya, our approach to generate recommendations is way more accurate than HITS. As HITS creates recommendations based solely on frequency, it is not suitable for such areas where users often visit diverse places and have a high rate of revisiting.

### 4.2.2. Recommendation for Existing Users

We investigated the efficiency of the recommendation list for existing users, and the experimental results are shown in Figures 9–12. Our LACF-Rec3 method demonstrated a significant improvement in terms of precision by 60.35%, 96.92%, 116.59%, and 185.49% when compared to U-CF-MS, TDCF, HITS, and UCF, respectively. Similarly, our method

also showed an improvement in terms of recall by approximately 61.44%, 98.39%, 93.23%, and 193.23% when compared to U-CF-MS, TDCF, HITS, and UCF, respectively.

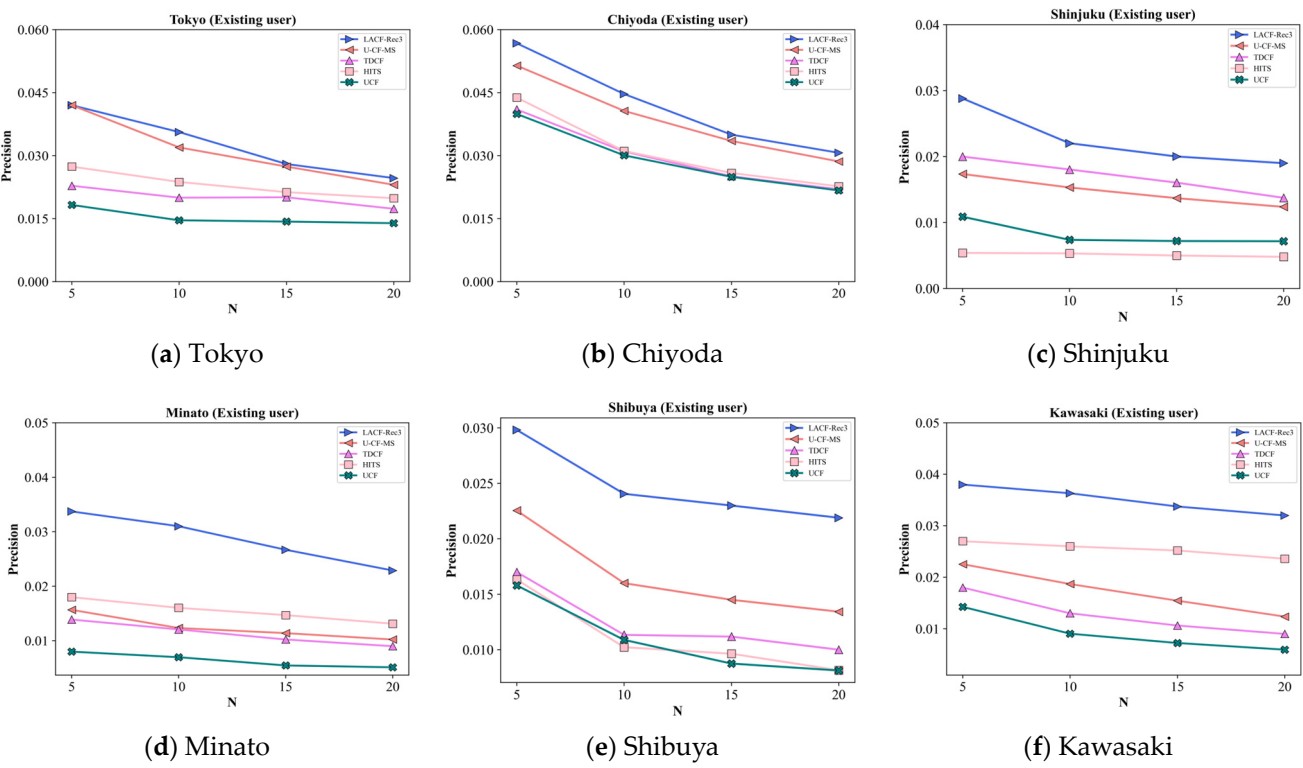

**Figure 9.** Precision metric for existing users with respect to the recommendation numbers.

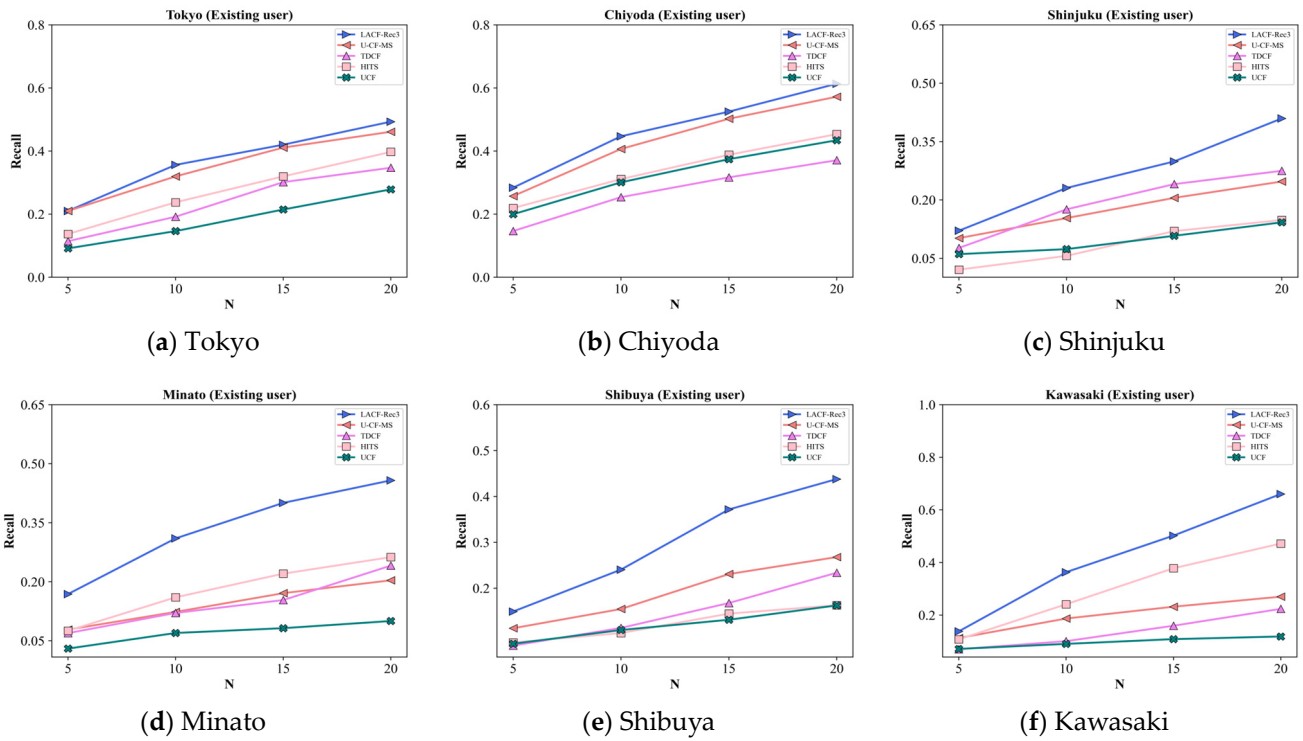

**Figure 10.** Recall metric for existing users with respect to the recommendation numbers.

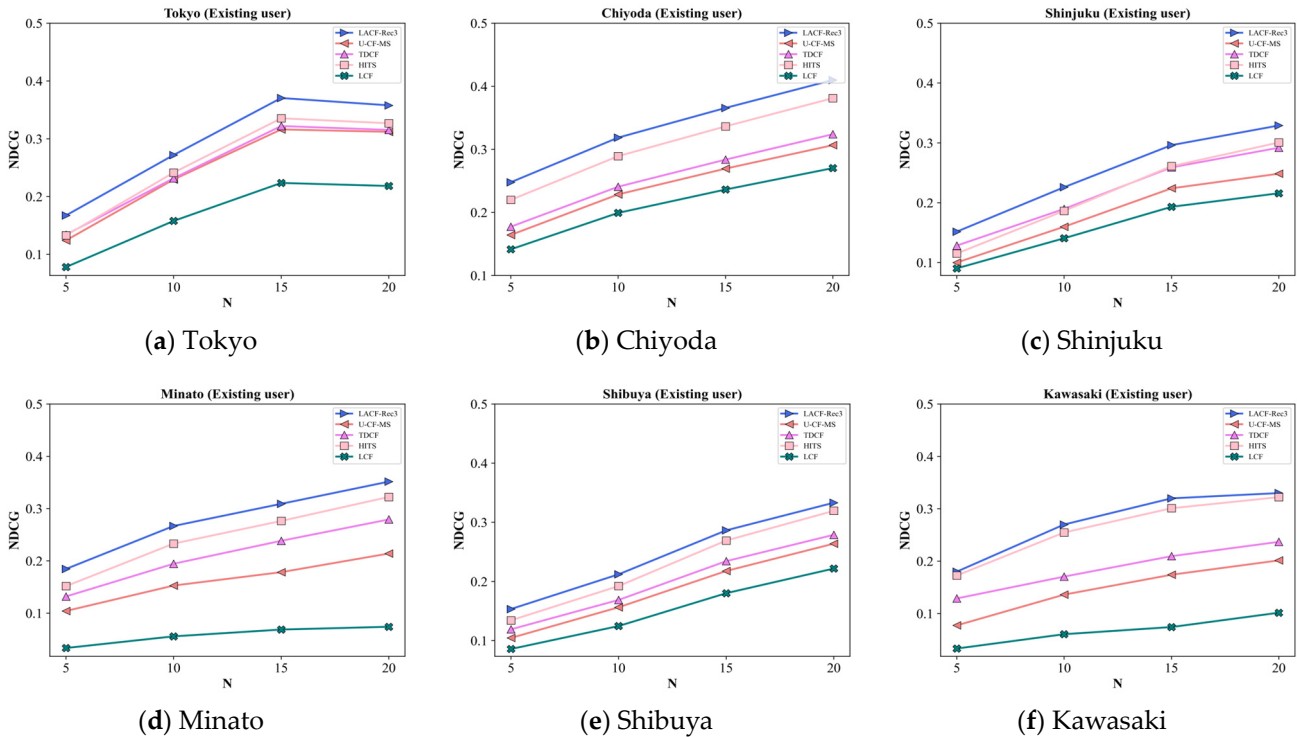

**Figure 11.** NDCG metric for existing users with respect to the recommendation numbers.

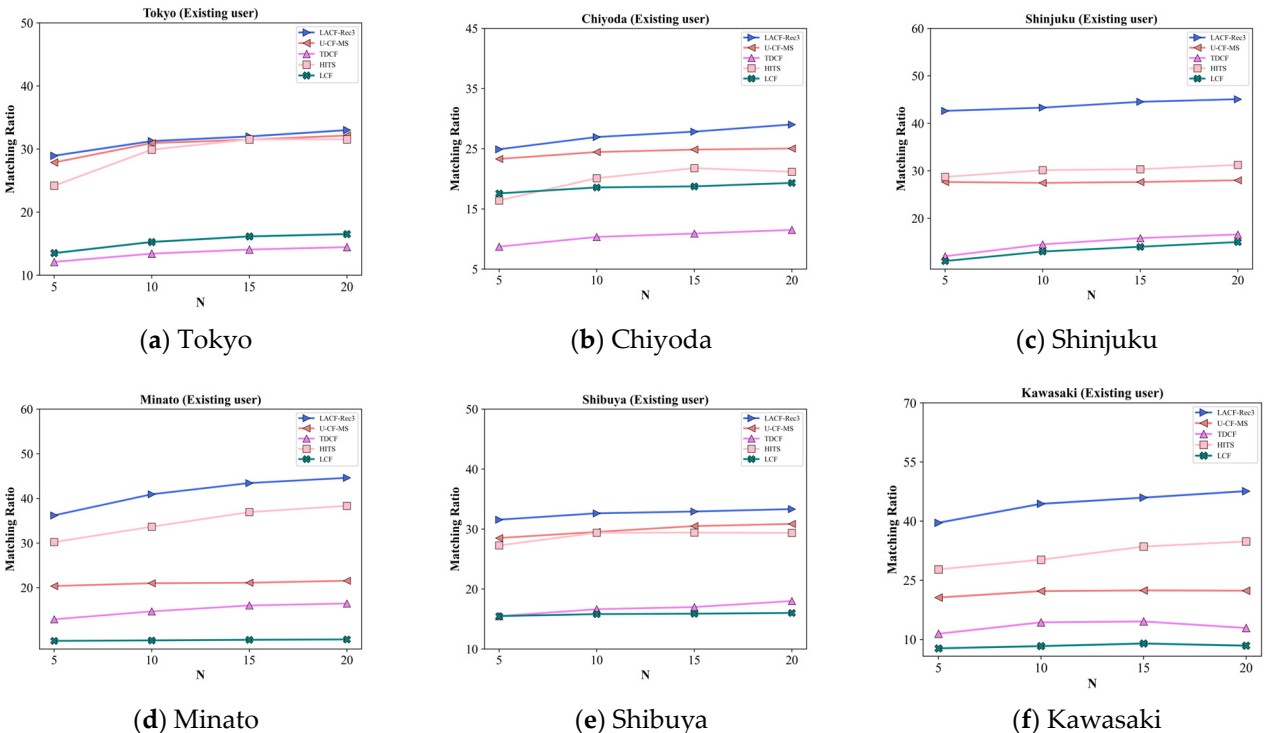

**Figure 12.** Matching ratio metric for existing users with respect to the recommendation numbers.

Since the target user has a check-in history, we can identify the target user's location interest by analyzing the distinctiveness of their favorite locations. Then, the distinctiveness of location associated with the target user can be compared with that of the other users, based on their check-in histories. Therefore, our proposed algorithm can identify users with similar interests to the target user more effectively than the other algorithms that directly consider only the locations that users have visited.

In addition, in real-world scenarios, users often have only a few check-ins at their favorite locations compared to a large number of locations. This data sparsity causes a difficulty in searching for other users who are similar to the target user. Our proposed method addresses this issue by recommending POIs based on the location distinctiveness related to relevant users, rather than comparing the list of locations visited by relevant users. As a consequence, our method is more capable of discovering relevant users with similar distinctiveness (and thus, similar preference) to the target user and is able to recommend lists of POIs that better align with the target users' preference.

Accordingly, the proposed method could generate more personalized POI recommendations than other algorithms, resulting in higher ranks of POIs in the user-selected recommendation list compared to the other methods. We observed that our proposed method outperformed U-CF-MS, TDCF, HITS, and UCF in terms of the NDCG metric by 46.49%, 26.17%, 9.36%, and 137.66%, respectively.

For the case of existing users, the evaluation of the matching ratio metric showed that our proposed algorithm provided better retrieval accuracy for POIs that matched the user preferences than that of U-CF-MS, TDCF, HITS, and UCF by 45.82%, 159.83%, 25.59%, and 205.21%, respectively. This is due to the fact that our method evaluates the distinctiveness by considering all three aspects, resulting in the ability to promote and recommend diverse types of POIs as well as avoid recommending popular yet common tourist destinations that could cause user disinterest.

Location recommendations may suffer from a tedium problem, where locations are chosen largely based on their popularity, and not on the variety and repetition rates. By considering the distinctiveness of locations based on the frequency, variety, and repetition, our POI recommendation list offers diverse and outstanding locations in each area. This helps users receive recommendations from multiple perspectives, thus solving the tedium problem.

Moreover, our analysis revealed that in areas with a large number of user check-ins (or a large number of existing users) such as Tokyo and Chiyoda, our LACF-Rec3 algorithm provided more accurate recommendations than those for areas with fewer user check-ins such as Shinjuku, Minato, Shibuya, and Kawasaki. More check-in data enabled our CF-3 to search for relevant users with similar check-in patterns more effectively. Therefore, our method excels in creating POI recommendation lists that align more accurately with the user preferences than the other methods.

In future work, we aim to focus on considering multiple aspects of distinctiveness among users and locations, as opposed to this work, where the distinctiveness of users and locations was based only on one aspect. This refinement becomes relevant in cases where users or locations exhibit multiple aspects of distinctiveness, leading to a possible decrease in the accuracy.

### 4.2.3. Privacy Issue Related to User Behavior Data Used in Recommendation Algorithms

The effectiveness of recommendation algorithms fundamentally relies on the acquisition and analysis of user behavior data such as the users' preferences, interests, or historical interactions to make personalized recommendations. Users are required to disclose information, creating a trade-off between utility and user privacy. While obtaining accurate recommendations is crucial, the sharing of personal information leads to the potential for privacy breaches, which may occur either deliberately (through snooping or hacking) or accidentally.

Although this study utilized a benchmark, which is masked and publicly available data, that did not contain sensitive information, it is essential to note that when the algorithm is deployed in production, users will need to provide information to access the desired features of our proposed method. Here, the privacy issue becomes a paramount concern. It is crucial to consider the legal and regulatory frameworks across all countries where the applications are utilized. Various countries have different data privacy regulations such as the European Union's General Data Protection Regulation (GDPR),

which imposes strict requirements on the collection, processing, and usage of user data. Non-compliance with these regulations can lead to substantial fines and legal repercussions. Obtaining informed consent from users for the collection and utilization of their data is also a critical legal obligation. To mitigate the potential privacy risks associated with the user behavior data, it is advisable to consider the adoption of existing privacy preserving recommendation methods such as cryptography-based approaches [37,38] or a two-stage privacy protection mechanism [39].

## 5. Conclusions

In this work, we proposed a novel hybrid POI recommendation system that combined link analysis and collaborative filtering, each of which was based on three visiting behaviors: frequency, variety, and repetition. Our method, called LACF-Rec3, focuses on the visiting characteristics of POIs and those of users to recommend POIs to existing and cold-start users. We introduced an extended version of the HITS-based model, called HITS-3, to generate top-ranked POIs for cold-start users. This model can handle not only the cold-start problem, but the sparsity problem by generating POI recommendation lists without using the user profile. To recommend interesting POIs for existing users, we introduced a novel collaborative filtering technique, called CF-3, which takes into consideration the distinctiveness of both the users (i.e., checked-in frequency, user variety, and checked-in repetitions) and POIs (checked-in frequency, user variety, and checked-in repetitions). CF-3 utilizes location distinctiveness, obtained from the HITS-3, to determine the similarity between the locations visited by target users and the ones visited by each relevant user. Then, we determined interesting locations based on the frequency of check-in, similarity of relevant users, and user distinctiveness to provide highly personalized recommendations. By considering distinctiveness, our method can help create diverse recommendation lists and address the tedium problem.

For our experimental evaluation on a real-world dataset, we used three well-known metrics: two for measuring accuracy (i.e., precision and recall) and the other for assessing the ranking accuracy (i.e., NDCG). In addition, this work proposed the matching ratio metric for evaluating the quality of recommendation lists by considering the distinctiveness of the target users in relation to the recommended POIs.

Our experimental results showed that our LACF-Rec3 method outperformed the baseline methods in terms of precision, recall, NDCG, and matching ratio. Our method effectively captured the diverse preferences of users by considering the distinctiveness of the POIs and users. For cold-start users, we recommended interesting locations based on location distinctiveness in their vicinity. As a result, our LACF-Rec3 method can generate diverse POI recommendation lists that match the preferences of target users.

For existing users, finding similarities in user behaviors can be facilitated by considering distinctiveness similarity rather than relying solely on location similarity. Furthermore, our method provides interesting locations to existing users based on their preferences.

The advantage of our LACF-Rec3 method lies in its ability to effectively capture the user preferences of target users, thereby generating POI recommendations. It is suitable for both cold-start users and existing users, relying on our proposed distinctiveness of locations and users, based on visiting frequency, variety, and repetition. Consequently, our method excels in the recommendation performance in terms of accuracy, ranking recommendation, and the alignment of recommended POIs with user preferences when compared to the baseline methods, which rely solely on frequency and/or repetition.

In future work, we plan to explore the characteristics of the POIs and users from multiple aspects (not just only one with the most distinctiveness). For example, some locations should be characterized by both the frequency and diversity of user check-ins, while some users may prefer visiting popular and frequently visited locations. In addition, we will attempt to evaluate the effectiveness of our method in terms of novelty and diversity. We also plan to incorporate novelty and diversity into the future version of our method to more efficiently cope with the tedium problem.

**Author Contributions:** Conceptualization, Sumet Darapisut, Komate Amphawan, Nutthanon Lee-lathakul and Sunisa Rimcharoen; Methodology, Komate Amphawan, Nutthanon Leelathakul and Sunisa Rimcharoen; Software, Sumet Darapisut; Validation, Sumet Darapisut and Komate Amphawan; Formal analysis, Komate Amphawan; Investigation, Komate Amphawan; Writing—original draft, Sumet Darapisut and Komate Amphawan; Writing—review & editing, Komate Amphawan, Nutthanon Leelathakul and Sunisa Rimcharoen; Visualization, Sumet Darapisut. All authors have read and agreed to the published version of the manuscript.

**Funding:** This research was funded by Faculty of Informatics, Burapha University, grant number 02/2565.

**Data Availability Statement:** Not applicable.

**Conflicts of Interest:** The authors declare no conflict of interest.

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
