# Peer review of "A Hybrid POI Recommendation System Combining Link Analysis and Collaborative Filtering Based on Various Visiting Behaviors"

_ijgi, doi:10.3390/ijgi12100431_

Round 1

Reviewer 1 Report

The manuscript proposed a novel POI recommendation system. It employs a hybrid approach of link analysis and collaborative filtering based on three visiting behaviors: frequency, variety, and repetition. However, some details still need to be discussed.

1、The paper is more like a list (such as definitions and experiments) than a systematic description of a method.

2、How does the proposed approach address data sparsity, cold start problem, and tedium problem? The article needs further detailed explanation.

3、What are the advantages of the LACF-Rec3 recommendation system compared to previous methods in terms of recommendation accuracy, ranking accuracy and matching rate?

4、A step-by-step example may help illustrate better your explanation.

5、If possible, it also advised to share a link to the source code or the executable code used to perform the experiments.

6、There are some structural problems in the paper, such as the pictures taking up too much space and the definition of formulas and symbols being unreasonable.

Can be improved

Author Response

Thank you very much for the consideration of our manuscript (ijgi-2613342) in the ISPRS International Journal of Geo-Information. We have revised our manuscript according to the Reviewer’s comments and responded to all comments from reviewer point by point. We have also reviewed again all sections of the paper and made it better in terms of presentation. Modified sentences have been marked in the highlighted version of the manuscript. We are appreciated your attentions and consideration of this revised version of the manuscript for publication in the ISPRS International Journal of Geo-Information.

Reviewer 2 Report

The paper proposes an algorithm for POI recommendation. Overall, the methodology and results are good.

In the introduction you mention some challenges of these methods. In my view you miss the most problematic challenges, the privacy issue. This type of methods require that you store behaviour information of the users which is, at least in my view, problematic (and in some countries possibly also not legal). This should at least be discussed in the paper.

In the paper you frequently use the terms: target users, existing users, and cold-start users. It would be good if you could define these user groups in the beginning. And what is really meant by an exiting user? Are the ones not belonging to this group non-existing? A definition is given in row 489-490, but this is rather late in the paper (and on row 490 I guess it should be "and" and not "or")

Section 2.3 is hard to follow before the details of your method LACF-Rec3 is provided. This section should be moved to the result section.

Detailed comments:

Abstract, second row from bottom: Tedious for whom?

Row 62 and 66: You need references to support theses claims.

Row 103: Here you should state that LACF-Rec3 consist of HITS-3 and CF-3.

Row 140: Here you use the term NDCG metrics which is not explained before the end of the paper.

Row 144: "existing literature" should be "current literature" 

Row 155: A->a

Row 167: Here you you explain the abbrevation "HITS (Hypertext Induced Topic Search)". This should have come earlier.

Row 242-244. Here you use "i.e." while I think it shoould be "e.g." (since you are only providing examples of e.g. context).

Figure 1: It is difficult to understand which part of the Figure that describes the Hits3 method and which part that describes the CF-3 method. Or is this figure only describing CF-3; if so, the figure caption should be changed.

Row 343: Should it be ISF_L here (and not ISF_U)?

Row 639: "famous" -> "common"

Have made a few comments on language above. I am not a native English speaker and do not have further comments. 

Author Response

(The authors gave the same response as above.)

Reviewer 3 Report

I think this paper proposes a new hybrid POI recommendation system, called LACF-Rec3, which combines different access behaviors based on link analysis and collaborative filtering, to provide personalized recommendations for location-based services. The research method and experimental results of the paper are very detailed, and the authors also compare and analyze the existing POI recommendation systems, and propose improvement schemes to improve the accuracy and diversity of the recommendations. In addition, the conclusion of the paper is very clear, and also provides some valuable suggestions for future research directions. Therefore, I think this paper is a very valuable research achievement, worthy of being published in an international journal.

However, the paper could provide more details about the specific techniques used in the link analysis and collaborative filtering methods. -Moreover, a more comprehensive evaluation, including comparison with other methods and other performance metrics, would strengthen the findings of the paper. -The authors should also discuss the limitations and potential future directions of the LACF-Rec3 system, and provide more insights on the data set used for the experiments.

Overall, the paper presents a novel hybrid POI recommendation system, which shows promising results, but some areas of further elaboration and evaluation would benefit it.  

Author Response

(The authors gave the same response as above.)

Round 2

Reviewer 1 Report

The revised version seems better than the older one.

Minor editing of the English language is required